# Diverse Community Data for Benchmarking Data Privacy Algorithms

**Aniruddha Sen**
University of Massachusetts
Amherst, MA 01003

**Christine Task**
Knexus Research
Oxon Hill, MD 20745
christine.task@knexusresearch.com

**Dhruv Kapur**
University of Michigan
Ann Arbor, MI 48109

**Gary Howarth**
NIST
Boulder, CO 80305
gary.howarth@nist.gov [*]

**Karan Bhagat**
Knexus Research
Oxon Hill, MD 20745

## Abstract

The Collaborative Research Cycle (CRC) is a National Institute of Standards and Technology (NIST) benchmarking program intended to strengthen understanding of tabular data deidentification technologies. Deidentification algorithms are vulnerable to the same bias and privacy issues that impact other data analytics and machine learning applications, and it can even amplify those issues by contaminating downstream applications. This paper summarizes four CRC contributions: theoretical work on the relationship between diverse populations and challenges for equitable deidentification; public benchmark data focused on diverse populations and challenging features; a comprehensive open source suite of evaluation metrology for deidentified datasets; and an archive of more than 450 deidentified data samples from a broad range of techniques. The initial set of evaluation results demonstrate the value of the CRC tools for investigations in this field.

## 1 Introduction

Deidentification algorithms take records linked to individuals and attempt to produce data that is dissociated from individuals but remains useful for analysis. Effective deidentification permits organizations to safely share useful information from potentially sensitive data. Such data can be used to train machine learning algorithms; expose fraud, waste, and abuse; improve health outcomes; and much more. Many approaches are available to deidentify data. Some approaches, such as statistical disclosure control, redact or suppress information that is deemed particularly identifying. Other approaches, such as synthetic data algorithms leverage generative models to reproduce sensitive data distributions using new, synthetic records. Differential privacy strengthens deidentification with a rigorous mathematical definition of privacy, producing synthetic data with quantifiable bounds on the influence of any real individual. While deidentification release algorithms may (but not necessarily) improve privacy, they can also distort data by introducing artifacts and bias. Identifying and resolving these issues is important, but it is not trivial to do.

Intuitively, individuals with many unusual values in the data are easier to identify than those with relatively common demographic characteristics. Consider data with two subpopulations of equal size: group A, who are internally homogeneous and have highly-correlated features; and group B, who are internally heterogeneous with relatively independent features. We can infer that group

---

[*] corresponding author

37th Conference on Neural Information Processing Systems (NeurIPS 2023) Track on Datasets and Benchmarks.

B will have more uniquely identifiable individuals. Heavy handed deidentification will eliminate or perturb group B to a greater extent than A, at the likely expense of reducing or meaningfully altering the representation of group B. We use *subpopulation dispersal* to refer to the sparsification of subpopulations, such as that caused by increased feature independence. We believe work on benchmarks containing subpopulation dispersal is imperative to overcoming these risks. Further, we advocate for evaluation metrics that explore the effects of subpopulation dispersal and other challenges induced by real-world data.

In this work we introduce the Collaborative Research Cycle (CRC) a benchmarking effort to compare deidentification methods hosted by the National Institute of Standards and Technology (NIST). The CRC includes target data, evaluation metrics, and a repository containing community-created deidentified data and their evaluation results. Our benchmark dataset contains examples of challenging, real-world conditions, such as subpopulation dispersal (Section 3.1). Together this program supports unprecedented rigorous exploration of deidentification algorithm behavior.

There are several existing synthetic data evaluation libraries, such as SDMetrics and YData (see Table 2 for a larger selection). These tools are focused on evaluating deidentified data with arbitrary schema and providing application-specific feedback. Our effort distinguishes itself by providing specific benchmarking data and a venue to contribute directly comparable samples of deidentification algorithm outputs and their evaluation results. We believe that shared benchmark data alongside common evaluation metrics promote understanding and exploration of a problem by providing common resources, vocabulary, and analytic framework. Thus, we have created the CRC as an arena to test, compare, and discuss varying approaches on equal footing.

We begin with a formal analysis of subpopulation dispersal. This concept underlies one source of tension between diversity, equity, and privacy that impacts all deidentification algorithms operating in real-world conditions. Specifically we show that subpopulations with greater feature independence leads to smaller cell counts in tabular data. We believe benchmark tabular data for deidentification technologies must present subpopulation dispersal to provide a realistically challenging target.

Next, we introduce a benchmark dataset of real demographic data, the NIST Diverse Communities Data Excerpts [1] (the Excerpts). The data are curated from the American Community Survey from the U.S. Census Bureau with 24 features and three geographic samples. We introduce a suite of benchmark software, the SDNist Deidentified Data Report Tool [2], which provides metrology and visualization functions comparing groundtruth and deidentified data. We use SDNist to demonstrate the presence of subpopulation dispersal within the Excerpts.

Finally, we introduce the CRC Data and Metrics Archive, a repository of crowd-sourced, deidentified instances of the Excerpts, all of which are benchmarked with SDNist. Members of industry, academia, and government have contributed more than 450 fully-evaluated entries to the repository, spanning many different deidentification approaches including differentially private techniques [3], cell suppression based $k$-anonymity[4], GAN-based synthetic data, and many others. We use SDNist and the archive to illustrate the impact of subpopulation dispersal on deidentification performance.

The CRC is a comprehensive effort to equip algorithm developers, data owners, and theoreticians the tools to evaluate and compare tabular deidentification technologies. All of the data (ground truth and deidentified instances) and evaluation software are open-sourced to allow transparent analysis. Further, the archived data has extensive, standardized metadata to facilitate parsing and comparing approaches, such as by feature subset, algorithm type, etc.

## 2  Distributional Diversity and Subpopulation Dispersal

We use the term *diverse populations* to refer to populations containing two or more subpopulations which differ from each other in terms of their feature correlations. By feature correlation we mean the predictive power of a feature value on other feature values for a given record. Intuitively, weaker feature correlations are more challenging for deidentification algorithms to model accurately. Here, we provide a robust formal analysis of these dynamics which helped inspire the Diverse Community Excerpts benchmark data.

Consider data represented in a histogram, with bins counting the number of occurrences of each record type (formally defined below). Individuals in bins with small counts are more uniquely identifiable–there are fewer people like them. Many privacy approaches focus on protecting these

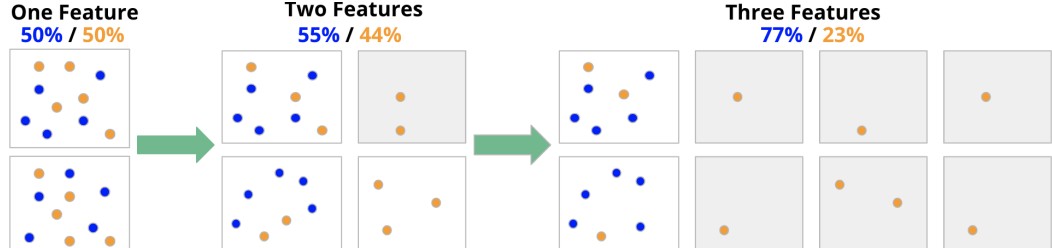

**One Feature**
**50% / 50%**

**Two Features**
**55% / 44%**

**Three Features**
**77% / 23%**

Figure 1: Boxes represent partition of observations by an increasing number of categorical features. Blue group features are highly correlated and orange group features are more independent. The orange group is, therefore, dispersed into small-count bins (in grey). The percentage label shows the relative representation of each group if small count bins were suppressed to protect privacy.

individuals. Traditional statistical disclosure control techniques like $k$-anonymity operate by perturbing or redacting records in small bins. Randomization approaches that rely on additive noise, such as differential privacy or subsampling, have much larger relative impact on these small bins. Non-differentially private synthetic data generators may have difficulty modeling sparser areas of the distribution. Thus, deidentified data fidelity is more challenging for individuals in small bins. However these individuals are not necessarily unimportant outliers. Depending on feature independence it is possible for a large subpopulation to become dispersed into small bins.

In diverse data the same schema may disperse one subpopulation and preserve the other, causing the first to potentially be erased by deidentification while the maintaining the second (see Figure 1). Therefore, it is vital to use diverse data to study algorithm behavior. We first introduce two relevant information theoretic tools and our notation. We then formally define *dispersal ratio* and derive bounds on the relationship between diversity and dispersal.

**Definition 2.1** (Entropy). The entropy of a discrete random variable $X$ is defined as:

$$\mathrm{H}(X) = -\sum_{x \in \mathcal{X}} p(x) \log p(x)$$

where $\mathcal{X}$ denotes the range of $X$ [5]. For this paper, we will be considering the empirical definition of entropy or observed entropy from the probability distribution corresponding to a histogram.

**Definition 2.2** (Uncertainty Coefficient). The uncertainty coefficient $U(X|F)$ is an information theoretic metric for quantifying the entropy ($H$) correlation between two random variables $X$ and $F$ (in our case an existing feature set $F$ and a new added feature $X$) [6]. It is a normalized form of mutual information. Note that $0 \leq U(X|F) \leq 1$. It is defined as:

$$U(X|F) = \frac{H(X) - H(X|F)}{H(X)}$$

An uncertainty coefficient of 1 implies that the random variable $X$ can be completely predicted by knowing the value of $F$, and an uncertainty coefficient of 0 implies that the random variable $X$ is independent of $F$. Thus, $U(X|F)$ inversely correlates with the independence of $X$ for some $F$. Note that this metric is asymmetric and may not be be equal if the variables are exchanged, i.e. $U(X|F) \neq U(F|X)$. This metric can be used only for categorical data, which makes it useful for a table-based partition schema where the data is distributed into cells with categorical feature labels.

Consider a population $P$ of individuals $i$ distributed in a table-based partitioned schema $S$. The proportion of the population $P$, with a given assignment of feature values corresponding to a bin in the schema $S$, is calculated as $p(bin_{S(i)}) = \frac{|\{i|bin_{S(i)}\}|}{|P|}$. Let the multivariate distribution of the initial feature set be described by $F$ in the schema $S$, and the univariate distribution of the new feature be described by $X$ in the schema $(S + X)$. The observed entropy $H(X)$ is dependent on the distribution of $X$, which remains the same over this operation for any added feature. We assume the size of the population is much larger than the number of bins, i.e., $|P| >> |Range(F + X)|$.

**Definition 2.3.** (Dispersal Ratio) Let the dispersal ratio for a population $P$ with the addition of feature $X$ be defined as:

$$Disperse(S, X, P) = |bin_{(S+X)(P)}| / |bin_{S(P)}|$$

where $bin_{S(P)}$ is defined as the set of all bins in the histogram corresponding to $P$ distributed in the schema $S$.

Real-world tabular survey data is often comprised primarily of non-ordinal, categorical features (multiple choice answers) that fall neatly into histogram bins. For a sub-population of a fixed size, definition 2.3 ratio captures the increase in the number of histogram bins for the population when a new feature is introduced. Thus, it captures the resulting reduction of the average count per bin and increases to the number of small-count bins. The lemma below provides some basic intuition behind the choice. The main result of this section is given in Theorem 2.3.

**Lemma 2.1.** An uncertainty coefficient of 1 is equivalent to a dispersal ratio of 1.
$$U(X|F) = 1 \iff Disperse(S, X, P) = 1$$

**Lemma 2.2.** An uncertainty coefficient of 0 leads to the maximum dispersal ratio.
$$U(X|F) = 0 \implies Disperse(S, X, P) = |Range(X)|$$

The above lemmas provide good intuition for our main argument in this section. Let's call these the trivial bounds for dispersal ratio on adding a feature $X$ to the schema. Moreover, we are also interested in quantifying how different patterns of feature correlations, represented by the uncertainty coefficient, impact the dispersal ratio for values within these bounds. The following theorem allows us to establish bounds on dispersal ratio as a function of the uncertainty coefficient or independence.

**Theorem 2.3.** Dispersal Ratio is bounded from above and below as function of the independence of the added feature as follows
$$\frac{|P| \cdot f(u)}{\log(|P|)|Range(F)|} \geq Disperse(S, X, P) \geq \frac{2^{f(u)}}{|Range(F)|}$$
where $f(u) := (1 - u)H(X) + H(F)$ with $u = U(X|F)$.

*Proof.* The following is a short sketch of the proof. The complete proof is detailed in Appendix C.2.

Let some arbitrary $u = U(X|F)$. Applying the well-known upper bound on entropy [5],
$$H(X, F) \leq \log_2(|Range(X, F)|) \tag{1}$$
provides a lower bound on dispersal ratio. Similarly, the observed entropy can be lower bounded by observing that each record in $S$ discretely contributes to the histogram. This provides an upper bound on the dispersal ratio. $\square$

Theorem 2.3 shows that, for some fixed value of entropy of the added feature, the non-trivial upper and lower bounds for the dispersal ratio decrease as the uncertainty coefficient increases, and vice-versa, according to the described behaviour of $f(u)$.

Now, we want to compare the effect of adding a new feature $X_1$ or $X_2$ to the schema. Let us assume that $X_1$ is more "independent" than $X_2$ of the distribution of $P$ in $S$ (note that this is equivalent to considering a single feature $X$ and two diverse subpopulations $P1, P2$ with differing relationships to $X$). We can then say that the uncertainty coefficient $u_1 = U(X_1|F)$ is lesser for $X_1$ as compared to $u_2$ corresponding to $X_2$. We can use our results from the above theorem to make this comparison as follows in Theorem 2.4. We define a couple of terms first for ease of notation.

Let the non-trivial lower bound for the dispersal ratio (>1) on adding feature $X$ be denoted as
$$LB(Disperse(S, X, P)) = \frac{2^{(1-u)H(X)+H(F)}}{|Range(F)|} \tag{2}$$

Let the non-trivial upper bound for the dispersal ratio (<$|Range(X)|$) on adding feature $X$ be denoted
$$UB(Disperse(S, X, P)) = \frac{|P| \cdot (1-u)H(X) + H(F)}{\log(|P|)|Range(F)|} \tag{3}$$

**Theorem 2.4.** Consider two features $X_1$ and $X_2$, identical in terms of entropy, that can be added to the schema. If $X_1$ has higher independence than $X_2$ with respect to $F$, it is equivalent to $X_1$ having a higher LB and higher UB for the dispersal ratio.
$$U(X_1|F) \leq U(X_2|F) \iff LB(Disperse(S, X_1, P) \geq LB(Disperse(S, X_2, P)$$
$$U(X_1|F) \leq U(X_2|F) \iff UB(Disperse(S, X_1, P) \geq UB(Disperse(S, X_2, P)$$

*Proof.* This follows from Theorem 2.3. The complete proof is detailed in Appendix C.2. $\square$

# 3 Introducing the Diverse Community Excerpts

The Excerpts are in the public domain and were designed to explore algorithm behavior on realistic data with diverse subpopulations (see Figure 2). We selected these data to be *tractable*, in light of a recurring problem identified in the NIST Differential Privacy Synthetic Data Challenge [7], NIST Differential Privacy Temporal Map Challenge, and the UNECE High-Level Working Group for the Modernisation of Statistics (HLG-MOS) Synthetic Data Test Drive [8], where the target data was too large or complex to identify, diagnose and address shortcomings in the deidendified data. This is a serious problem; consumers of deidentified government data cannot afford to overlook even subtle introductions of bias, artifacts, or privacy leaks. But data properties like subpopulation dispersal can induce defects that are not visible in aggregate utility metrics used by most privacy researchers and data competitions. And deeper exploration is intractable when considering hundreds of features and millions of records. Addressing these issues requires tools designed to make them accessible.

The Excerpts consist of a curated geography and feature set derived from the significantly larger 2019 American Community Survey (ACS) Public Use Microdata Sample (PUMS) [9], a product of the U.S. Census Bureau. The Census Bureau applies privacy measures to the data[10], and no independent privacy risks of this subset of the data. The Excerpts serve as benchmark data for two currently active, open source projects at the National Institute of Standards and Technology (NIST)–SDNist Deidentified Data Report Tool and the 2023 Collaborative Research Cycle (CRC).

The Excerpts' feature set was developed with input from U.S. Census Bureau experts in adaptive sampling design (see [11]). To identify a small set of communities with challenging, diverse distributions, the Excerpts leverage previous work on geographical differences in CART-modeled synthetic data (see [12, Appendix B]). The open source SDNist library which accompanies the excerpts was developed with input from HLG-MOS participants and synthetic data contractors working with the U.S. Census. In this section we provide an overview of the Excerpts; in the next section we demonstrate their ability to identify and diagnose problems in deidentification algorithms.

## 3.1 Data Overview

**Feature Selection** The original ACS schema contains over four hundred features, which poses difficulties for diagnosing shortcomings in deidentification algorithms. The Excerpts use a small but representative selection of 24 features, covering major census categories: demographic, household and family, geographic, financial, work and education, disability, and survey weights (discussed in the 'Challenges' below). Several Excerpts' features were not in the original ACS features and were designed to provide easy access to certain information. Population DENSITY allows models to distinguish rural and urban geographies, INDP_CAT aggregates industry codes into categories, PINCP_DECILE aggregates incomes into percentile bins relative to the record's state, and EDU simplifies school-

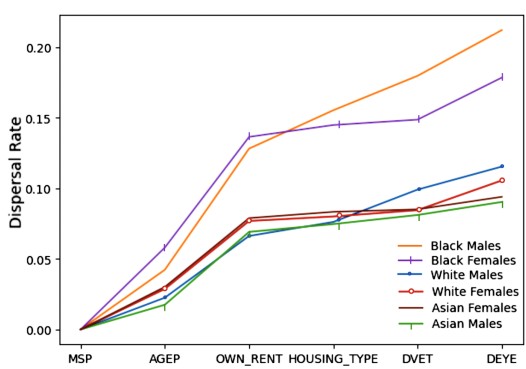

Figure 2: Effect of feature addition to dispersal rate of the Excerpts data (National sample).

ing to focus on milestone grades and degrees. The Excerpts' repository has detailed information including several recommended feature subsets for use in diverse subpopulation analyses.

**Recommended Feature Subsets** To support algorithms that require smaller schema and explore the impact of different types and combinations of features, we provide recommended subsets of the 24 features (see Appendix D). Algorithm performance may differ significantly between feature subsets (see Appendix E.4).

**Subpopulation Dispersal** Figure 2 shows subgroup dispersal by race in the Excerpts data; as new features are added that are more independent for one group and more tightly correlated for other groups, the independent group becomes dispersed across the features space. The experiments in Section 5.1 use the CRC archive to demonstrate the impact of this on algorithm performance for the largest four race + sex subgroups. However, we note there's nothing 'magic' about this choice of demographic subgroups, other population partitions exhibit similar behavior (e.g., partitions by

sex + disability, employment sector). A rigorous multidimensional understanding of the relationship between variation in patterns/strength of feature correlations, variations in distribution density, and pitfalls for deidentification is the future research this benchmarking program is designed to support.

**Three Excerpts Data Sets** Data sets of differing sizes, reflecting different types of communities, induce different algorithm behavior. To prevent overfitting research conclusions to a particular type of community, the Excerpts contains three target data sets. The "Massachusetts" data contains 7634 records drawn from five Public Use Microdata Areas (PUMAs) of demographically homogeneous communities from the North Shore to the west of the greater Boston area; the "Texas" data with 9276 records is drawn from six racially diverse PUMAs of communities surrounding Dallas-Fort Worth, Texas area. The "National" data with 27254 records is drawn from 20 diverse PUMAs across the United States; it is an expansion of a challenging PUMA set used during the design of the 2020 Decennial Census Disclosure Avoidance System algorithm [12]. The experiments in this paper use the National Excerpts data.

**Metadata and Documentation:** The Diverse Community Excerpts include JSON data dictionaries with complete feature definitions, a github readme with detailed usage guidance, and illustrative "postcard" introductions to the real-world communities in the data.

**Challenges:** The Excerpts have been designed to be representative of real-world survey data conditions. In addition to diversity, these include other challenging properties: logical constraints between features (e.g., AGE = 6 places a constraint on MARITAL STATUS) that can be difficult for synthetic data generators (see Figure 6). Additional modeling challenges include heterogeneous feature types and cardinalities (e.g., DEYE has two categorical values while INDUSTRY has 271, INCOME is an integer while POVERTY INCOME RATIO is continuous). Uneven feature granularity can amplify problems with unequal subpopulation dispersal (ACS 2019 RAC1P uses one code for Asian, but four detailed codes for Native American). We also include the sampling weights that survey data users need to simulate a full population; these lose their original meaning after deidentification changes the data sample, and this is a largely unsolved problem.

**Limitations:** Although ACS data consumers generally assume the data remains representative of the real population, the ACS PUMS data has had basic statistical disclosure control deidentification applied (as noted above), which may impact its distribution. Additionally, there are shortcomings in the Diverse Community Excerpts that we plan to address in future versions: the Excerpts do not currently include the ACS Household IDs (which would support joining individuals in the same households for social network synthesis), Individual IDs (relevant for reidentification research), or a clear training/testing partition (important for differential privacy research).

## 4   Introducing the SDNist Deidentified Data Report Tool

The SDNist Deidentified Data Report Tool is a Python library that is undergoing active development. SDNist evaluates fidelity, utility, and privacy of a given set of deidentified instances of the Excerpts and generates human- and machine-readable summary quality reports enumerated and illustrated for each utility and privacy metric. It is comprised largely of fidelity metrics drawn from data stakeholders through the HLG-MOS Synthetic Data Test Drive, and aims to provide a comprehensive view of algorithm behavior. Examples of complete reports, including detailed metric definitions, are linked in the Appendix E. A metric overview is below; usage is demonstrated in Section 5.

**Univariate and Correlation Metrics** The SDNist report covers univariate feature distributions as well as pairwise feature correlations with Pearsons and Kendall Tau Coefficients.

**Higher Order Similarity Metrics ($k$-marginal, equivalent subsample)** Data analysts often use more than two features. SDNist provides higher order distributional similarity comparison using the *k-marginal* edit distance similarity score [7, 12, 13]. This compares the target data and deidentified data densities across $k$-marginal queries, with a maximum score of 1000 indicating identical data sets. The report provides overall and and per-PUMA 3-marginal scores. To aid interpretation, the report includes an *Equivalent Subsample (ES)* table that provides a comparison to edit distance induced by sampling error. A $k$-marginal score with an ES of 15% is about as dissimilar from the target data as a data set created by uniformly randomly discarding 85% of the target data.

**Task Comparison Metrics (propensity, linear regression)** The report includes two machine learning task-based metrics. The *propensity* metric (Figure 6) trains a classifier to predict whether a record

belongs to the target or deidentified data. The classifier's per-record prediction confidence (propensity distribution) is plotted for both data sets; when the classifier finds it impossible to distinguish the two, the traces will match with a spike at 50 %. The *linear regression* metric focuses on a task very relevant to the subpopulation dispersal problem (see Figure 6). It uses linear regression to summarize the relationship between educational attainment and income decile for race + sex subgroups. The target regression line is given in red, the deidentified line is in green. Correlation strength differs for different subgroups; this causes dispersal and introduces challenges for deidentification. Some deidentification approaches artificially weaken the correlation, and others strengthen it.

**Visual Diagnosis Metrics (PCA, regression heatmap)** To help diagnose the causes of low similarity scores, and identify bias/artifacts introduced during deidentification, the SDNist report includes several visualizations. *Pairwise Principle Component Analysis (PCA)* supports direct comparison of the target and deidentified data distribution using scatterplots across principle component axes. An interactive PCA exploration tool is available. In Figure 6, the target data (ground truth) PCA scatterplot is shown on the left in blue, the deidentified data plot is shown on the right, in green. For additional insight, records with marital status = N/A (indicating children < 15) are highlighted in red in both plots. Algorithms that provide good privacy and utility will reproduce the shape of the target scatterplot using new points (i.e., deidentified records). The *linear regression heatmaps* (Figure 5) provide another resource for identifying artifacts. The target data distribution is shown as a red-blue heatmap of distribution densities (normalized by educational attainment level). The deidentified heatmaps show deviation from the target distribution: brown indicates the deidentified data contains too many individuals in that category, purple indicates it contains too few. Artifacts and bias are identifiable as large ares of blue or brown blocks, indicating where privacy techniques systematically erase or over emphasize parts of the distribution.

**Empirical Privacy Metrics (Unique Exact Match)** Privacy protection can be considered empirically, or formally. For techniques with parameterized guarantees (such as differential privacy or $k$-anonyimity) we include parameter values in their metadata. However, any technique (including DP) may have catastrophic privacy failures. The *Unique Exact Match (UEM)* metric is a measure of unambiguous privacy leakage applicable to all deidentification techniques; it simply counts the percentage of unique individuals in the target data who remain present unaltered in the deidentified data. Being unique, these individuals are more vulnerable to reidentification; being unaltered, this reidentification may not be difficult. As Table 1 (and Appendix E.3) shows, a simple differentially private hisotgram technique with weak guarantee $\epsilon = 10$ can reproduce its input data nearly exactly without violating DP. The UEM metric allows us to identify when one technique is performing very badly on privacy, or identify when one technique is performing significantly worse than another. Because UEM doesn't test non-trivial reidentification attacks, it cannot be used to determine whether a technique provides good privacy, or which of two similarly performing techniques is best. We plan to expand the SDNist privacy metrics; however, this starting point provides valuable insights.

## 5 The Collaborative Research Cycle Data and Metrics Archive

The CRC Data and Metrics Archive is comprised of deidentified data samples generated by applying deidentification algorithms to the Diverse Community Excerpts, using varied techniques, feature subsets, and parameters. It opened collection in March 2023 and is still actively accepting submissions; at time of writing it includes 453 samples of 33 techniques from 18 deidentification libraries. Below we provide a brief overview of its contents, based on meta-data available in the archive index file.

**Privacy Types.** We delineate three basic types: 56 samples use Statistical Disclosure Control Techniques that use perturbation or redaction to anonymize the target data while leaving it substantially intact; 139 are non-differentially private synthetic data techniques which generate new records to fit the target distribution; and 258 are Differentially Private techniques which add randomization to satisfy a formally private guarantee limiting an individual record's influence on the relative probability of possible outputs. [3]

**Algorithm Types.** To support meta-analysis, we provide a high-level categorization of approaches. All approaches can be implemented with or without formal privacy guarantees. In the archive, 166 deidentified samples use neural network modeling approaches, 143 fit traditional statistical models, 67 iteratively update a naive distribution to mimic query results on the target data, 56 use statistical disclosure control anonymization, 12 are simple histogram techniques that alter counts of record occurrences, and nine select new points using geometric interpolation between target records.

**Exploring Diverse Subgroup Dispersal in the Archive.** In Section 2 and Figure 1 we discussed the impact subpopulation dispersal and resulting small count cells can have on performance after deidentification. In the analyses below we look at how this impact plays out in practice in the archive. These deidentified samples used the National Excerpts target data and demographic-focused feature set. To compare subgroup utility, we use the $k$-marginal metric across 4-marginals including the race + sex features. Only samples with 15% subsample equivalence or greater are included.

Figure 3 shows the distribution of $k$-marginal scores for the four largest race + sex subpopulations; the more dispersed group in general has poorer performance, but some deidentified data samples show good performance overall. Understanding what distinguishes deidentification algorithms that are more or less impacted by subgroup dispersal is a key goal of this benchmarking project, and identifying them is a first step. Figure 4a provides further breakdown, showing how subgroup performance varies with overall data fidelity. We see that the expected difference in performance appears regularly but not constantly, some algorithms overcome it and provide similar performance for all groups. Figure 4b supplements this with privacy leak information using the UME metric (Section 4). Some high performing algorithms are actually identi-

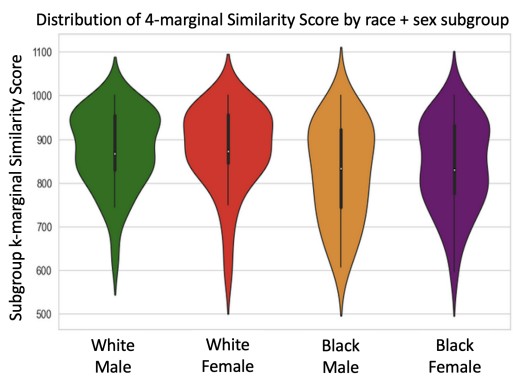

Figure 3: 4-marginal edit distance similarity, by demographic subgroup, from the CRC Data and Metrics Archive

cally reproducing the target data, providing very little privacy, while others are able to maintain distributional fidelity for all subgroups using substantially new records. The CRC benchmarking program is designed to support the research that leads to a formal understanding of this behavior, and the development of high fidelity, equitable deidentification techniques with negligible privacy leaks. The detailed metrics for all samples appearing in these charts are provided in Appendix E2; no current archive algorithm fully satisfies all three requirements for this feature set.

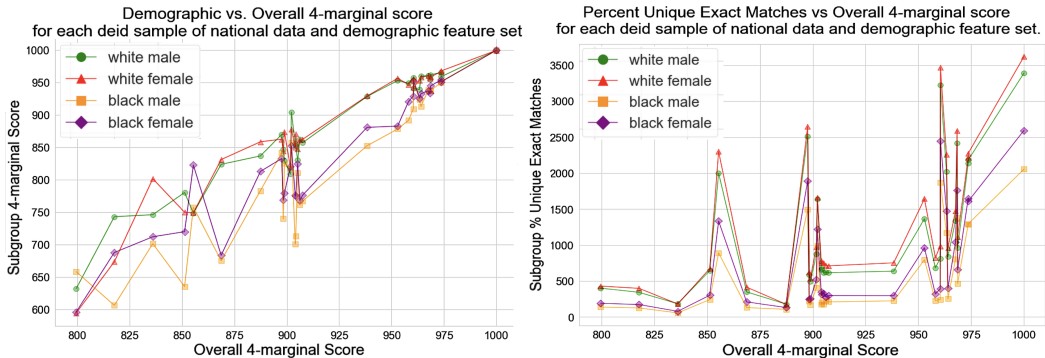

(a) $k$-marginal similarity performance.   (b) UEM privacy performance.

Figure 4: Subgroups utility/privacy performance, ordered by total population 4-marginal performance.

## 5.1 Exploration of Selected Algorithms

Table 1 uses seven interesting Archive contributions to demonstrate the Excerpts' efficacy as a tool for exploring and understanding the behavior of deidentification algorithms.

The regression metric (Figure 5) allows us to explore the impact of variable subpopulation dispersal. Very poor performing algorithms show little impact (ADSGAN, PACSynth), while others have worse performance on the more dispersed subpopulation (black women). CART and MST perform well, but introduce a slight bias that strengthens the correlation between high educational attainment and high income in the dispersed group (brown artifact in upper right of heatmap). This could obscure the real disparity between the groups.

| Library and Algorithm | Privacy Type | Algorithm Type | Priv. Leak (UEM) | Utility (ES) |
|---|---|---|---|---|
| DP Histogram ($\epsilon = 10$) | differential privacy (DP) | simple histogram | 100% | $\sim 90\%$ (988) |
| R synthpop CART model [14] | non-DP synthetic data | multiple imputation decision tree | 2.54% | $\sim 40\%$ (935) |
| MOSTLY AI SDG [15] [16] | non-DP synthetic data | proprietary pre-trained neural network | 0.03% | $\sim 30\%$ (921) |
| SmartNoise MST ($\epsilon = 10$) [17] | DP | probabilistic graphical model (PGM) | 13.6% | $= 10\%$ (969) |
| SDV CTGAN [18] [19] | non-DP synthetic data | generative adversarial network (GAN) | 0.0% | $\sim 5\%$ (775) |
| SmartNoise PACSynth ($\epsilon = 10$) [20] | DP + $k$-anonymity | constraint satisfaction | 0.87% | $\sim 1\%$ (551) |
| synthcity ADSGAN [21] [22] | custom noise injection | GAN | 0.0% | $< 1\%$ (121) |

Table 1: **Performance of selected deidentification algorithms,** see Appendix E.1 for additional details.

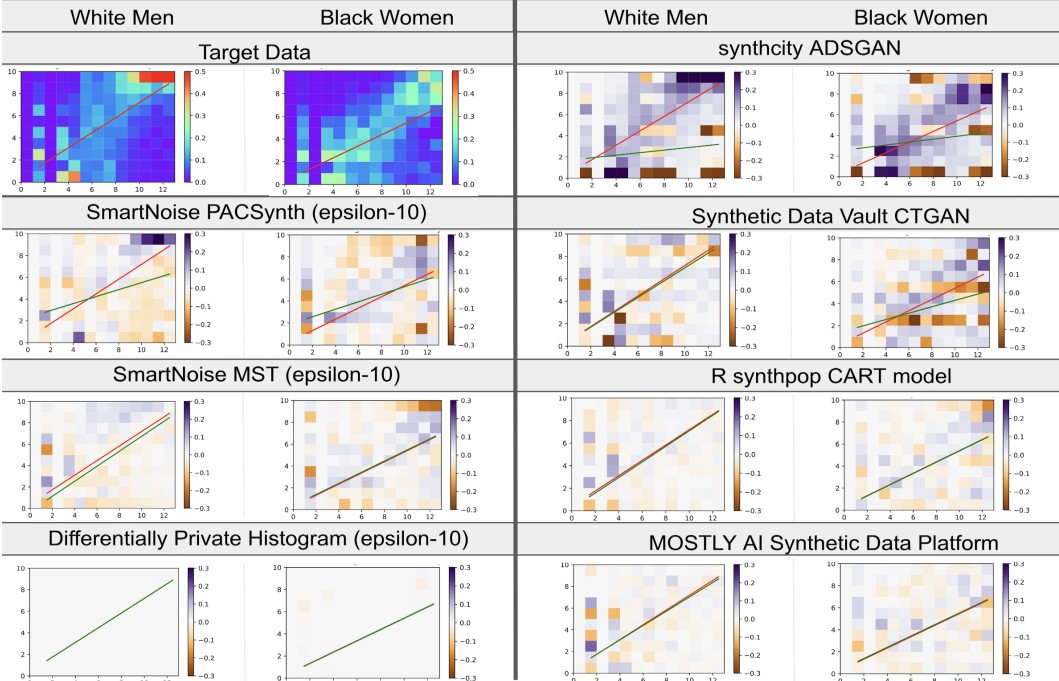

Figure 5: Linear regression metric showing how well the selected algorithms maintain the relationship between educational attainment ($x$). and income decile ($y$) for different demographic groups.

Considering privacy, all algorithms in the left column guarantee differential privacy with an identical privacy parameter setting $\epsilon$=10 (weak privacy protection). However we see in Table 1 that techniques with the same formal guarantee can produce widely varied results in terms of both privacy and utility. PACSynth has additional $k$-anonymity protection which eliminates rare, dispersed records; this provides good privacy, but has much poorer utility with unusual impacts on the distribution that can be identified in this and other metrics. Meanwhile, the simple DP Histogram provides almost no privacy protection at all, reproducing the target data nearly exactly. MST is possibly good compromise of privacy and utility (and would provide better privacy at smaller $\epsilon$), but non-differentially private techniques CART and MostlyAI have significantly less trivial privacy leakage (by UEM metric, Section 4) and better utility.

In figure 6 we present two more SDNist metrics: The PCA metric uses scatterplots along principle component axes to compare the shapes of distributions. The propensity metric trains a classifier to distinguish between real and synthetic data; if the data are indistinguishable the propensity distributions will peak at the center (indicating the classifier can only make a '50/50 guess' whether a given record is from the target data or deidentified data). If the deidentified data contains artifacts or bias these will be visible as mismatches in the shape of the scatterplot. This figure compares the three neural network synthesizers in our selection, and provides further explanation for the utility values noted in Table 1. CTGAN is not preserving constraints that hold for child records (highlighted in

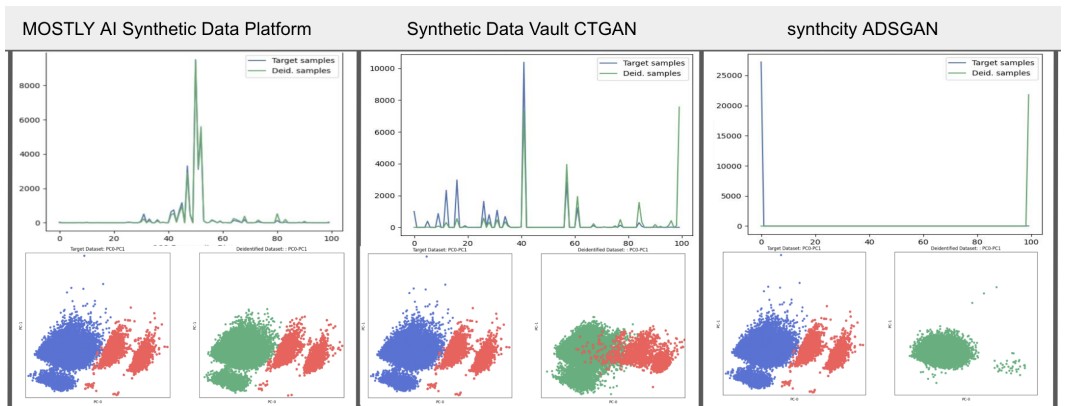

Figure 6: Propensity and PCA metrics.

red), and is missing a constraint on housing features (visible as the three well-separated clusters in the target data, but blurred together in the CTGAN data). Examples of records violating constraints might be a 7-year-old widow or an inmate who owns her jail; these can be confidently classified as synthetic by the propensity metric. Meanwhile, the ADSGAN data contains no children at all, and has retained none of the structure of the target data. Note that these results are more informative than simple edit distance utility scores. We are able to identify and diagnose specific algorithm behaviors induced by the real world challenges in the Excerpt data.

| Tool name | Univariate / Correlations | Higher Order Simililarity | Task Comparison | Visual Diagnosis | Privacy analysis | Built-in synthesizer(s) |
|---|---|---|---|---|---|---|
| Anonymeter[23] | | | | | ✓ | |
| Data Responsibly[24] | ✓ | ✓ | | | | ✓ |
| NHS Synthetic Data Pipeline | ✓ | ✓ | ✓ | ✓ | ✓ | ✓* |
| OpenDP[25] | ✓ | | | | | ✓* |
| R Synthpop[14] | ✓ | ✓ | ✓ | | ✓ | ✓ |
| SDGym / SDMetrics[18] | ✓ | ✓ | ✓ | | ✓ | ✓ |
| **SDNist** | ✓ | ✓ | ✓ | ✓ | ✓ | |
| Synthcity[22] | ✓ | ✓ | ✓ | | ✓ | ✓ |
| Table Evaluator | ✓ | ✓ | ✓ | ✓ | | |
| TAPAS | | | | | ✓ | |
| YData | ✓ | ✓ | ✓ | ✓ | | ✓ |

Table 2: Synthetic data evaluation tools

# 6   Related Works

Table 2 provides a broad survey of comparable tools for benchmarking and evaluating synthetic data, including both library links and research citations. Libraries annotated with '*' are specifically configured to evaluate data from their built-in synthesizer models. Visual diagnosis refers to tools that visualize and explore data within a high dimensional space (e.g., principal component analysis). Privacy analysis specifically refers to comparison of output records to input records (e.g., re-identification, replication). Many of these tools have built in data and all of them also have 'bring-your-own-data' capabilities. SDNist, and the CRC program in general, distinguish themselves with a more extensive evaluation set, and a specific goal to promote the state of research in the deidentification field as a whole, aiming at rigorously well understood high performance on diverse, real-world data. Unlike the CRC, none of these resources are designed to focus energy on improving synthetic data specifically by comparing algorithm performance on unified data with common metrics.

# 7   Conclusion

Rather than focus on evaluating arbitrary data, we present tools that are tailored to a specific dataset. We are motivated by previous evaluations of differentially private [26] and of GAN-based synthetic data [27] generators that show surprising differences in algorithm performance. The Excerpts, partnered with SDNist and theory presented here, provide a compact vehicle to address diverse subpopulations, which we show is a persistent problem facing deidentification technologies.

**Disclaimer.** The Collaborative Research Cycle (CRC), the Data and Metrics Archive, and SDNist are intended as tools to encourage investigation and discussion of deidentification algorithms, and they are not intended or suitable for product evaluation. The National Institute of Standards and Technology does not endorse any algorithm included in these resources. No mention of a commercial product in this paper or any CRC resource constitutes an endorsement.

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
