# Supplemental Information for "Diverse Community Data for Benchmarking Data Privacy Algorithms"

October 27, 2023

## Supplemental Information Contents

# A   Disclaimer

The Collaborative Research Cycle (CRC), the Data and Metrics Archive, and SDNist are intended as tools to encourage investigation and discussion of deidentification algorithms, and they are not intended or suitable for product evaluation. The National Institute of Standards and Technology does not endorse any algorithm included in these resources. No mention of a commercial product in this paper or any CRC resource constitutes an endorsement.

# B  Dataset Provisions

## B.1  Dataset URLs

- The NIST public data repository access point.

- Direct data access link.

- Direct data access to the data dictionary.

- The dataset DOI is 10.18434/mds2-2895.

### B.1.1  Dataset Format Notes

The raw data are in comma-sperated value (CSV) format with (Javascript object notation) JSON data dictionaries defining valid values.

The NIST data repository has a structured metadata retrieval system that interfaces with data.gov and conforms to FAIR principles and the best practice for Federal Data Strategy. See additional information here.

## B.2  Author Statement

The authors bear all responsibility in case of violation of rights. We have confirmed licensing and provide detailed information in the article and in the dataset datasheet.

## B.3  Hosting and Licensing

The data associated with this publication were created, hosted, and maintained, by the National Institute of Standards and Technology in their permanent data repository, in perpetuity.

The data are in the public domain. NIST statement on software and data:

"NIST-developed software is provided by NIST as a public service. You may use, copy, and distribute copies of the software in any medium, provided that you keep intact this entire notice. You may improve, modify, and create derivative works of the software or any portion of the software, and you may copy and distribute such modifications or works. Modified works should carry a notice stating that you changed the software and should note the date and nature of any such change. Please explicitly acknowledge the National Institute of Standards and Technology as the source of the software. NIST-developed software is expressly provided "AS IS." NIST MAKES NO WARRANTY OF ANY KIND, EXPRESS, IMPLIED, IN FACT, OR ARISING BY OPERATION OF LAW, INCLUDING, WITHOUT LIMITATION, THE IMPLIED WARRANTY OF MERCHANTABILITY, FITNESS FOR A PARTICULAR PURPOSE, NON-INFRINGEMENT, AND DATA ACCURACY. NIST NEITHER REPRESENTS NOR WARRANTS THAT THE OPERATION OF THE SOFTWARE WILL BE UNINTERRUPTED OR ERROR-FREE, OR THAT ANY DEFECTS WILL BE

CORRECTED. NIST DOES NOT WARRANT OR MAKE ANY REPRESEN-
TATIONS REGARDING THE USE OF THE SOFTWARE OR THE RESULTS
THEREOF, INCLUDING BUT NOT LIMITED TO THE CORRECTNESS,
ACCURACY, RELIABILITY, OR USEFULNESS OF THE SOFTWARE. You
are solely responsible for determining the appropriateness of using and distribut-
ing the software and you assume all risks associated with its use, including but
not limited to the risks and costs of program errors, compliance with applicable
laws, damage to or loss of data, programs or equipment, and the unavailability
or interruption of operation. This software is not intended to be used in any
situation where a failure could cause risk of injury or damage to property. The
software developed by NIST employees is not subject to copyright protection
within the United States."

# C Datasheet for dataset "NIST Diverse Communities Data Excerpts"

Questions from the Datasheets for Datasets paper, v7.

Jump to section:

- Motivation

- Composition

- Collection process

- Preprocessing/cleaning/labeling

- Uses

- Distribution

- Maintenance

## C.1 Motivation

### C.1.1 For what purpose was the dataset created?

The NIST Diverse Communities Data Excerpts (the Excerpts) are demographic data created as benchmark data for deidentification technologies.

The Excerpts are designed to contain sufficient complexity to be challenging to de-identify and with a compact feature set to make them tractable for analysis. We also demonstrate the data contain subpopulations with varying levels of feature independence, which leads to small cell counts, a particularly challenging deidentification problem.

The Excerpts serve as benchmark data for two open source projects at the National Institute of Standards and Technology (NIST): the SDNist Deidentified Data Report tool and the 2023 Collaborative Research Cycle (CRC).

### C.1.2 Who created the dataset (e.g., which team, research group) and on behalf of which entity (e.g., company, institution, organization)?

The Excerpts were created by the Privacy Engineering Program of the Information Technology Laboratory at the National Institute of Standards and Technology (NIST).

The underlying data was published by the U.S. Census Bureau as part of the 2019 American Community Survey (ACS) Public Use Microdata Sample (PUMS).

### C.1.3 Who funded the creation of the dataset?

The data were collected by the U.S. Census Bureau, and the Excerpts were curated by NIST. Both are U.S. Government agencies within the Department of Commerce. Aspects of the Excerpts creation were supported under NIST contract 1333ND18DNB630011.

### C.1.4 Any other comments?

No.

## C.2 Composition

### C.2.1 What do the instances that comprise the dataset represent (e.g., documents, photos, people, countries)?

The instances in the data represent individual people.

The Excerpts consist of a small curated geography and feature set derived from the significantly larger 2019 American Community Survey (ACS) Public Use Microdata Sample (PUMS), a publicly available product of the U.S. Census Bureau. The original ACS schema contains over four hundred features, which poses difficulties for accurately diagnosing shortcomings in deidentification algorithms. The Excerpts use a small but representative selection of 24 features, covering major census categories: Demographic, Household and Family, Geographic, Financial, Work and Education, Disability, and Survey Weights. Several Excerpts features are derivatives of the original ACS features, designed to provide easier access to certain information (such as income decile or population density).

There is only one type of instance. All records in the data represent separate, individual people.

### C.2.2 How many instances are there in total (of each type, if appropriate)?

There are three geographic partitions in the data. See the "postcards" and data dictionaries in each respective directory for more detailed information. Instances in partitions:

- `national`: 27254 records

- `massachusetts`: 7634 records

- `texas`: 9276 records

### C.2.3 Does the dataset contain all possible instances or is it a sample (not necessarily random) of instances from a larger set?

The data set is a curated sample of the ACS by geography, with a reduced feature set designed to provide a tractable foundation for benchmarking deiden-

tification algorithms (24 features rather than the original ACS's 400 features). Geographically it is comprised of 31 Public Use Microdata Areas

- `national`: 27254 records drawn from 20 Public Use Microdata Areas (PUMAs) from across the United States. This excerpt was selected to include communities with very diverse subpopulation distributions.

- `texas`: 9276 records drawn from six PUMAs of communities surrounding Dallas-Fort Worth, Texas area. This excerpt was selected to focus on areas with moderate diversity.

- `massachusetts`: 7634 records drawn from five PUMAs of communities from the North Shore to the west of the greater Boston, Massachusetts area. This excerpt was selected to focus on areas with less diversity.

### C.2.4 What data do each instance consist of?

The instances are individual, tabular data records in CSV format with Demographic, Household and Family, Geographic, Financial, Work and Education, Disability, and Survey Weights features.

In addition, there are metadata and documentation, schema files for each of the three geographies containing the features and valid data ranges in JSON format, and 'postcard' documentation with English-language descriptions of the areas described by the data in PDF format. There are also an overarching readme and data dictionary.

### C.2.5 Is there a label or target associated with each instance?

No. These data are not designed specifically for classifier tasks.

### C.2.6 Is any information missing from individual instances?

There is no missing information in these excerpts, all records are complete.

### C.2.7 Are relationships between individual instances made explicit (e.g., users' movie ratings, social network links)?

Relationships between records have not been included in this version of the data. Although the Excerpts data do contain multiple individuals from the same household, it does not include the ACS PUMS Household ID or relationship features needed to join them into a network. We expect to include those features in a future update to the Excerpts.

### C.2.8 Are there recommended data splits (e.g., training, development/validation, testing)?

There are three geographic partitions to facilitate benchmarking algorithms on populations with differing levels of heterogeneity/diversity (MA, TX and

National). There are no splits designed specifically for training and testing purposes. All of the data presented at this time are from the 2019 ACS collection. In the future we plan to add additional years.

### C.2.9  Are there any errors, sources of noise, or redundancies in the dataset?

Although ACS data consumers generally assume the data remains representative of the real population, the ACS PUMS data have had basic statistical disclosure control deidentification applied (including swapping and subsampling), which may impact its distribution. For more information, see documentation from the U.S. Census Bureau.

### C.2.10  Is the dataset self-contained, or does it link to or otherwise rely on external resources (e.g., websites, tweets, other datasets)?

All of the data are self-contained within the repository. The data are drawn from public domain sources and, thus, have no restrictions on usage.

### C.2.11  Does the dataset contain data that might be considered confidential?

The Excerpts are a subset public data published by the U.S. Census Bureau. The U.S. Census Bureau is bound by law, under Title 13 of the U.S. Code, to protect the identities of individuals represented by the data. See here for details on the Census' data stewardship. The Census takes elaborate steps to reduce risk of re-identification of individuals surveyed and provide information regarding their suppression scheme here.

### C.2.12  Does the dataset contain data that, if viewed directly, might be offensive, insulting, threatening, or might otherwise cause anxiety?

No.

### C.2.13  Does the dataset relate to people?

Yes.

### C.2.14  Does the dataset identify any subpopulations (e.g., by age, gender)?

The data include demographic features such as Age, Sex, Race and Hispanic Origin which may be used to disaggregate by subpopulation. It additionally includes non-demographic features such as Educational Attainment, Income

Decile and Industry Category which also produce subpopulation distributions with disparate patterns of feature correlations.

The racial and ethnicity subpopulation breakdown by geography is as follows (note that Hispanic origin and race are separate features):

- `MA Dataset (less diverse)`: 4% Hispanic and 89% White, 7% Asian, 2% Black, 2% Other, 0% AIANNH

- `TX Dataset (more diverse)`: 19% Hispanic and 85% White, 7% Black, 4% Other, 3% Asian, 1% AIANNH

- `National Dataset (especially diverse)`: 10% Hispanic and 56% White, 22% Black, 10% Other, 9% Asian, 3% AIANNH

### C.2.15  Is it possible to identify individuals (i.e., one or more natural persons), either directly or indirectly (i.e., in combination with other data) from the dataset?

The Excerpts are survey results from real individuals as collected by the U.S. Census Bureau. See response above for more information about Census' data protections.

The subset of the Census' data that we provide here introduces no additional information, and, therefore, does not increase the risk of identifying individuals.

### C.2.16  Does the dataset contain data that might be considered sensitive in any way (e.g., data that reveals racial or ethnic origins, sexual orientations, religious beliefs, political opinions or union memberships, or locations; financial or health data; biometric or genetic data; forms of government identification, such as social security numbers; criminal history)?

Yes. These data are detailed demographic records. See response above for more information about Census Bureau's data protections.

### C.2.17  Any other comments?

No.

## C.3  Collection process

### C.3.1  How was the data associated with each instance acquired?

These data is a curated geographic subsample of the 2019 American Community Survey Public Use Microdata files. The U.S. Census Bureau details its survey data collection approach here.

### C.3.2 What mechanisms or procedures were used to collect the data (e.g., hardware apparatus or sensor, manual human curation, software program, software API)?

See previous response.

### C.3.3 If the dataset is a sample from a larger set, what was the sampling strategy (e.g., deterministic, probabilistic with specific sampling probabilities)?

The data set is a (deterministic) curated sample by geography. It is comprised of 31 Public Use Microdata Areas

- `national`: 27254 records drawn from 20 Public Use Microdata Areas (PUMAs) from across the United States. This excerpt was selected to include communities with very diverse subpopulation distributions.

- `texas`: 9276 records drawn from six PUMAs of communities surrounding Dallas-Fort Worth, Texas area. This excerpt was selected to focus on areas with moderate diversity.

- `massachusetts`: 7634 records drawn from five PUMAs of communities from the North Shore to the west of the greater Boston, Massachusetts area. This excerpt was selected to focus on areas with less diversity.

### C.3.4 Who was involved in the data collection process (e.g., students, crowdworkers, contractors) and how were they compensated (e.g., how much were crowdworkers paid)?

[See response above.]

### C.3.5 Over what timeframe was the data collected?

These data was collected during 2019.

### C.3.6 Were any ethical review processes conducted (e.g., by an institutional review board)?

The Excerpts are a curated subsample of existing public data published by the U.S. Government. No internal review board (IRB)review was necessary by institution policy.

### C.3.7 Does the dataset relate to people?

Yes.

### C.3.8 Did you collect the data from the individuals in question directly, or obtain it via third parties or other sources (e.g., websites)?

Other Sources. These data are a curated geographic subsample of the 2019 American Community Survey Public Use Microdata files, which are available here.

### C.3.9 Were the individuals in question notified about the data collection?

Yes. See response above.

### C.3.10 Did the individuals in question consent to the collection and use of their data?

Yes. See response above.

### C.3.11 If consent was obtained, were the consenting individuals provided with a mechanism to revoke their consent in the future or for certain uses?

See response above.

### C.3.12 Has an analysis of the potential impact of the dataset and its use on data subjects (e.g., a data protection impact analysis) been conducted?

These data is a curated geographic subsample of the 2019 American Community Survey (ACS) Public Use Microdata files. Many investigations have examined ACS data with some information published by the Census Bureau itself.

The data presented here, the Excerpts, are a subset of the data and present no additional risks to the subjects surveyed by the Census.

### C.3.13 Any other comments?

No.

## C.4 Preprocessing/cleaning/labeling

### C.4.1 Was any preprocessing/cleaning/labeling of the data done (e.g., discretization or bucketing, tokenization, part-of-speech tagging, SIFT feature extraction, removal of instances, processing of missing values)?

The original ACS data are clean, and no class labeling was done. However, several Excerpts features are new derivatives of ACS features designed to provide easier access to certain information. Population DENSITY divides PUMA

population by surface area and allows models to distinguish rural and urban geographies. INDP_CAT aggregates detailed industry codes into a small set of broad categories. PINCP_DECILE aggregates incomes into percentile bins relative to the record's state. And, EDU simplifies the original ACS schooling feature to focus on milestone grades and degrees.

### C.4.2 Was the "raw" data saved in addition to the preprocessed/cleaned/labeled data (e.g., to support unanticipated future uses)?

See the U.S. Census Bureau's documentation for information about published ACS data.

### C.4.3 Is the software used to preprocess/clean/label the instances available?

The preprocessing was minimal (addition of a small set of derivative features), and can be reproduced as described above. The code is not currently available.

### C.4.4 Any other comments?

No.

## C.5 Uses

### C.5.1 Has the dataset been used for any tasks already?

The Excerpts serve as benchmark data for two open source projects at the National Institute of Standards and Technology (NIST): the SDNist Deidentified Data Report tool and the 2023 Collaborative Research Cycle (CRC).

### C.5.2 Is there a repository that links to any or all papers or systems that use the dataset?

No. Users are not mandated to contribute their work to any central repository. We publish user-contributed data here. We recommend that data users cite our work using the dataset DOI.

### C.5.3 What (other) tasks could the dataset be used for?

The Excerpts were designed for benchmarking privacy-preserving data deidentification techniques such as synthetic data or statistical disclosure limitation. However, they can be used to study the behavior of any tabular data machine learning or analysis technique when applied to diverse populations. Synthetic data generators are just an especially verbose application of machine learning (producing full records rather than class labels), so tools designed to improve understanding of synthetic data have potential for a much broader application.

### C.5.4 Is there anything about the composition of the dataset or the way it was collected and preprocessed/cleaned/labeled that might impact future uses?

The U.S. Census Bureau recommends using sampling weights to account for survey undersampling and generate equitable full population statistics. The PWGPT feature included in the Excerpts is the person (record) level sampling weight. For full population statistics, each record should be multiplied by its sampling weight.

### C.5.5 Are there tasks for which the dataset should not be used?

The Excerpts are suitable for any application relevant to government survey data over the selected feature set.

### C.5.6 Any other comments?

No.

## C.6 Distribution

### C.6.1 Will the dataset be distributed to third parties outside of the entity (e.g., company, institution, organization) on behalf of which the dataset was created?

### C.6.2 Has the dataset been used for any tasks already?

The Excerpts serve as benchmark data for two open source projects at the National Institute of Standards and Technology (NIST): the SDNist Deidentified Data Report tool and the 2023 Collaborative Research Cycle (CRC).

### C.6.3 How will the dataset will be distributed (e.g., tarball on website, API, GitHub)?

10.18434/mds2-289

### C.6.4 When will the dataset be distributed?

The dataset is currently available to the public.

### C.6.5 Will the dataset be distributed under a copyright or other intellectual property (IP) license, and/or under applicable terms of use (ToU)?

The data are in the public domain. See the following statement from NIST.

### C.6.6 Have any third parties imposed IP-based or other restrictions on the data associated with the instances?

No. All data are drawn from public domain sources.

### C.6.7 Do any export controls or other regulatory restrictions apply to the dataset or to individual instances?

No. All data are drawn from public domain sources and have no known export or regulatory restrictions.

### C.6.8 Any other comments?

No.

## C.7 Maintenance

### C.7.1 Who is supporting/hosting/maintaining the dataset?

This dataset is hosted by NIST and maintained by the Privacy Engineering Program.

### C.7.2 How can the owner/curator/manager of the dataset be contacted (e.g., email address)?

Dataset managers can be reached by raising an issue, emailing the Privacy Engineering Program, or by contacting the project principal investigator, Gary Howarth.

### C.7.3 Is there an erratum?

There have been small updates to the meta-data data dictionary.json files (for example, to improve clarity in descriptive strings for features). The data are maintained in a public GIt repository and, thus, all changes to the data are recorded in a public ledger.

### C.7.4 Will the dataset be updated (e.g., to correct labeling errors, add new instances, delete instances)?

Since the data are excerpts from the 2019 release of the American Community Survey, we do not expect any updates to labels or instances. We do plan on one mayor updated version release in the future with the following improvements:

- `Household ID features`: Allows joins between individuals in the same household

- `Individual ID`: Supports reidentification research.

- `Training Data Partition`: Including excerpts from 2018 for algorithm development/training and as a baseline for reidentification studies

- `Large-sized low-diversity excerpt`: Our current low-diversity excerpts, MA and TX, have much fewer records than our high-diversity excerpt, National; this can be a confounding factor for comparative analyses.

### C.7.5     If the dataset relates to people, are there applicable limits on the retention of the data associated with the instances (e.g., were individuals in question told that their data would be retained for a fixed period of time and then deleted)?

These data are in the public domain and as such there are no retention limits.

### C.7.6     Will older versions of the dataset continue to be supported/hosted/maintained?

The data are maintained in a public Git repository and, thus, all changes to the data are recorded in a public ledger. There are specific releases in the repository that capture major data milestones.

### C.7.7     If others want to extend/augment/build on/contribute to the dataset, is there a mechanism for them to do so?

We invite the public to use and build on these resources. First, these resources are provided by NIST as a public service, and the public is free to integrate these resources into their own work. Second, we invite the public to raise issues in the dataset repository, allowing for a transparent interaction. Individuals and groups wishing to make substantial contributions are encouraged to contact the project principal investigator, Gary Howarth.

### C.7.8     Any other comments?

No.

# D  Math Appendix

## D.1  Proofs of Lemmas 2.1 and 2.2

We introduced the concept of dispersal ratio in the main paper with the purpose of a giving the reader a clear and intuitive explanation of the term. In doing so, we omitted some formal results that might be interesting to examine in order to understand the mechanics behind dispersal ratio and independence. The perceptive reader may have noticed that we stated two lemmas in Section 2 without proving them. Recall the definition of dispersal ratio.

**Definition D.1** (Dispersal Ratio). Let the dispersal ratio for a population $P$ with the addition of feature $X$ be defined as

$$Disperse(S, X, P) = |bin_{(S+X)(P)}|/|bin_{S(P)}|$$

We begin by providing proofs of Lemma 2.1 (corresponding to Lemma C.1) and Lemma 2.2 (corresponding to Lemma C.2) as stated in Section 2. We follow it up with a result that may be of interest in Section C.3. These proofs follows the same framework and terminology as used in the main paper.

**Lemma D.1.** An uncertainty coefficient of 1 is equivalent to a dispersal ratio of 1.
$$U(X|F) = 1 \iff Disperse(S, X, P) = 1$$

*Proof.*
$$U(X|F) = 1 \Rightarrow \frac{H(X) - H(X|F)}{H(X)} = 1 \Rightarrow H(X|F) = 0$$

Consider the following result. $H(X|F) = 0$ if and only if $X$ is a function of $F$ i.e., $\forall f : p(f) > 0$, there is only one possible value of $x$ with $p(x, f) > 0$ [1].

Let the function $g$ between $X$ and $F$ be denoted by $F = g(X)$. Applying the result here, let there be $m$ elements in the domain of $F$, which implies there can be no more than $m$ elements in the co-domain of $X$, to constitute a valid function. Let the elements in the range of $F$ be denoted by $f_1, f_2...f_m$, and that of $X$ be denoted by $x_1, x_2...x_{m'}$.

Since $X$ is a function of $F$, there exists only one element $x_i \in X$ corresponding to $f_j \in F$.

Thus, all bins in the schema $(S + x)$ can be denoted by $(f_i, x_i) = (f_i, g(f_i))$. Since there are $m$ bins, corresponding the size of the domain, in $F$, there will be exactly $m$ bins in $F' = (F, X)$. Therefore,

$$|bin_{(S+X)(P)}| = |bin_{S(P)}|$$

$$\Rightarrow Disperse(S, X, P) = 1$$

Similarly, the converse of the lemma can be proved by taking the converse of the above result and considering that the inverse of the function $g' = g^{-1}(x)$ for $x$: $(F, X) \to F$ is uniquely defined if the dispersal ratio is 1. $\square$

**Lemma D.2.** An uncertainty coefficient of 0 leads to the maximum dispersal ratio.

$$U(X|F) = 0 \implies Disperse(S, X, P) = |Range(X)|$$

*Proof.*

$$U(X|F) = 0 \implies \frac{H(X) - H(X|F)}{H(X)} = 0 \implies H(X|F) = H(X)$$

This implies $X$ and $F$ are independent observations [1]. Observe that the range of $Y = (X, F)$ can take maximum $n_{max} = |Range(X)||Range(F)|$ values since it is the number of elements in $X \times F$. Note that $|Range(F)| = |bin_{S(P)}|$. Here, $Range(Y) = n_{max}$ due to the independence of $X$ and $F$ since

$$\forall x, f : Pr[Y = y] = (Pr[X = x] * Pr[F = f]) \neq 0$$

As there are $n_{max}$ non-zero values for the probability distribution of $Y$, the size of the range of $Y$ is maximum. Note that $Y$ exactly expresses the distribution of values in the schema $(S + X)$.
Therefore,

$$|bin_{(S+X)(P)}| = |bin_{S(P)}||Range(X)|$$
$$\implies Disperse(S, X, P) = |Range(X)|$$

which is the maximum dispersal ratio since $|bin_{S(P)}| * |Range(X)|$ was maximized.

$\square$

## D.2 Proofs of Theorems 2.3 and 2.4

Here, we provide the detailed proofs of Theorem 2.3 (corresponding to Lemma C.3) and Theorem 2.4 (corresponding to Lemma C.4) as stated in Section 2. These proofs follows the same framework and terminology as used in the main paper.

**Theorem D.3.** Dispersal Ratio is bounded from above and below as function of the independence of the added feature as follows

$$\frac{|P| \cdot f(u)}{\log(|P|)|Range(F)|} \geq Disperse(S, X, P) \geq \frac{2^{f(u)}}{|Range(F)|}$$

where $f(u) := (1 - u)H(X) + H(F)$ with $u = U(X|F)$.

*Proof.* Lemma 2.1 and Lemma 2.2 give an upper and lower bound for the dispersal ratio which is

$$1 \leq Disperse(S, X, P) \leq |Range(X)|$$

corresponding to

$$1 \geq U(X|F) \geq 0$$

Independence is quantified through the uncertainty coefficient $U(X|F)$. Recall that as $U$ decreases, independence increases, and vice-versa.

Let some arbitrary $u = U(X|F)$. From the definition of $U(X|F)$,

$$u = \frac{H(X) - H(X|F)}{H(X)} \Rightarrow H(X|F) = (1-u)H(X)$$

Rewriting in terms of the joint entropy [1] $H(X, F)$,

$$H(X, F) = (1-u)H(X) + H(F) \tag{1}$$

Applying a well-known upper bound on entropy [1], and substituting in equation (1),

$$H(X, F) \leq \log_2(|Range(X, F)|) \implies |Range(X, F)| \geq 2^{(1-u)H(X)+H(F)} \tag{2}$$

Following from our definition of dispersal ratio,

$$Disperse(S, X, P) = \frac{|bin_{(S+X)(P)}|}{|bin_{(S)(P)}|} = \frac{|Range(X, F)|}{|Range(F)|} \tag{3}$$

Thus,

$$Disperse(S, X, P) \geq \frac{2^{(1-u)H(X)+H(F)}}{|Range(F)|} \tag{4}$$

Now, from the definition of entropy,

$$\mathrm{H}(X, F) = -\sum_{x \in (\mathcal{X} \times \mathcal{F})} p(x) \log p(x)$$

Since each bin, corresponding to $x$ in the above equation, must have at least one person in order to contribute to the entropy, $p(x) \geq \frac{1}{|P|}$ where $|P|$ is the size of the population. To analyze any arbitrary distribution, we can first allocate one person to each bin in the range, by definition of range. Now, we have to distribute $|P| - |Range(X, F)|$ people in $|Range(X, F)|$ bins. Entropy is minimized when all the rest of the people are put in one bin. This distribution gives a lower bound for entropy in terms of the range of $(X, F)$. Formally, we get the following inequality,

$$H(X, F) \geq (|Range(X, F)| - 1)\left[\frac{\log(|P|)}{|P|}\right] + \frac{|P| - (|Range(X,F)| - 1)}{|P|} \log\left(\frac{|P|}{|P| - (|Range(X,F)| - 1)}\right) \tag{5}$$

Since $|P| \gg |Range(X, F)| \gg 1$, the second term is neglected as $1 \cdot \log(\frac{|P|}{|P|}) = 0$,

$$\implies \mathrm{H}(X, F) \geq |Range(X, F)|\left[\frac{\log(|P|)}{|P|}\right]$$

Substituting in equation (1) and rearranging terms, we get

$$Disperse(S, X, P) \leq \frac{|P| \cdot ((1-u)H(X) + H(F))}{\log(|P|)|Range(F)|} \tag{6}$$

We can improve on our trivial bounds of 1 and $|Range(X)|$, from Lemma 2.1 and Lemma 2.2, for the dispersal ratio corresponding to a given $u$. Combining these two results with the improved upper and lower bounds from equation (3) and equation (5),

$$\min\left\{\frac{|P|\cdot((1-u)H(X)+H(F))}{\log(|P|)|Range(F)|}, |Range(X)|\right\} \geq Disperse(S, X, P) \geq \max\left\{\frac{2^{(1-u)H(X)+H(F)}}{|Range(F)|}, 1\right\}$$
(7)

From equation (1), we can write $H(X, F)$ as a function of $u$ i.e., $f(u) := (1-u)H(X)+H(F)$. Also note that as $u$ increases, $f(u)$ decreases and vice-versa. This gives a simpler form for our above equation (also considering only non-trivial bounds),

$$\frac{|P|\cdot f(u)}{\log(|P|)|Range(F)|} \geq Disperse(S, X, P) \geq \frac{2^{f(u)}}{|Range(F)|}$$
(8)

$\square$

This result shows that, for some fixed value of entropy of the added feature, the non-trivial upper and lower bounds for the dispersal ratio decrease as the uncertainty coefficient increases, and vice-versa, according to the described behaviour of $f(u)$.

Now, we want to compare the effect of adding a new feature $X_1$ or $X_2$ to the schema. Let us assume that $X_1$ is more "independent" than $X_2$ of the distribution of $P$ in $S$ (note that this is equivalent to considering a single feature $X$ and two diverse subpopulations $P1, P2$ with differing relationships to $X$). We can then say that the uncertainty coefficient $u_1 = U(X_1|F)$ is lesser for $X_1$ as compared to $u_2$ corresponding to $X_2$. We can use our results from the above theorem to make this comparison as follows in Theorem 2.4. We define a couple of terms first for ease of notation.

Let the non-trivial lower bound for the dispersal ratio ($>1$) on adding feature $X$ be denoted as

$$LB(Disperse(S, X, P)) = \frac{2^{(1-u)H(X)+H(F)}}{|Range(F)|}$$
(9)

Let the non-trivial upper bound for the dispersal ratio ($<|Range(X)|$) on adding feature $X$ be denoted

$$UB(Disperse(S, X, P)) = \frac{|P|\cdot(1-u)H(X)+H(F)}{\log(|P|)|Range(F)|}$$
(10)

**Theorem D.4.** Consider two features $X_1$ and $X_2$, identical in terms of entropy, that can be added to the schema. If $X_1$ has higher independence than $X_2$ with respect to $F$, it is equivalent to $X_1$ having a higher LB and higher UB for the dispersal ratio.

$$U(X_1|F) \leq U(X_2|F) \iff LB(Disperse(S, X_1, P) \geq LB(Disperse(S, X_2, P)$$

$$U(X_1|F) \leq U(X_2|F) \iff UB(Disperse(S, X_1, P) \geq UB(Disperse(S, X_2, P)$$

*Proof.* Consider a population $P$ distributed in a table-based partitioned schema $S$. Note that higher independence corresponds to a lower uncertainty coefficient. Let $u_1 = U(X_1|F)$, $u_2 = U(X_2|F)$ and $H(X) = H(X_1) = H(X_2)$. The following result can be derived from Theorem 2.3. Let us consider the lower bound first. From equation (9),

$$\frac{LB((Disperse(S, X_1, P))}{LB((Disperse(S, X_2, P))} = \frac{\frac{2^{(1-u_1)H(X)+H(F)}}{|Range(F)|}}{\frac{2^{(1-u_2)H(X)+H(F)}}{|Range(F)|}} \implies \frac{LB((Disperse(S, X_1, P))}{LB((Disperse(S, X_2, P))} = 2^{H(X)(u_2-u_1)}$$

Since entropy is always greater than or equal to $0$ , $H(X) \geq 0$ [1]. Thus,

$$LB(Disperse(S, X_1, P) \geq LB(Disperse(S, X_2, P) \iff u_2 - u_1 \geq 0 \iff u_1 \leq u_2$$

Similarly for upper bound, from equation (10), we get

$$\frac{UB((Disperse(S, X_1, P))}{UB((Disperse(S, X_2, P))} = \frac{(1-u_1)H(X) + H(F)}{(1-u_2)H(X) + H(F)}$$

Clearly,

$$UB(Disperse(S, X_1, P) \geq UB(Disperse(S, X_2, P) \iff (1-u_1)H(X)+H(F) \geq (1-u_2)H(X)+H(F)$$

$$\iff u_1 \leq u_2$$

$\square$

## D.3    Additional Material

We now show an interesting consequence of the relation between dispersal ratio and the initial population. The following lemmas prove that a small population size can lead to small cell counts.

Consider a population $P$ with a sub-population $P_1$, distributed in a table-based partitioned schema. Consider an individual $i \in P_1$, who gets placed in a bin under schema $S$. We denote the size of that bin as $size(bin_{S(i)})$. Let a feature $f$ be added to the schema.

**Lemma D.5.** The dispersal ratio is always greater than or equal to 1.

$$Disperse(S, f, P_1) \geq 1$$

*Proof.* Consider an arbitrary $bin_S(i)$ in the schema $S$ with the $m$ features in the feature set $f_1, f_2, f_3...f_m$.

Adding a new feature $f$ to the schema $S$ with feature values (say) in the set $V = \{v_1, v_2\}$ will subdivide all records in $f_1, f_2, f_3...f_m$ into $f_1, f_2, f_3...f_m, v_1$ and $f_1, f_2, f_3...f_m, v_2$, by the definition of partitioning.

$bin_S(i)$ in the schema $S$ will be replaced by at least one bin or more, in the schema $(S+f)$. Thus, the dispersal ratio for the sub-population of $bin_{S(i)} : i \in P_1$ is always greater than 1.

Since for each disjoint sub-population corresponding to each bin $\in S$, this ratio is greater than one, the dispersal ratio for the overall population $P_1$ over the schema $S$ and adding a new feature $f$, is also greater than 1. $\qquad\square$

**Definition D.2** (Average bin size for population $P_1$). It is defined as

$$\left[ \sum_{S(i):i\in P_1} size(bin_{S(i)}) \right] / |bin_{S(P_1)}|$$

**Lemma D.6.** If a new feature f is added to the schema denoted by $S+f$, then the average bin size will stay the same or decrease.

$$\left[ \sum_{S(i):i\in P_1} size(bin_{S(i)}) \right] / |bin_{S(P_1)}| \geq \left[ \sum_{(S+f)(i):i\in P_1} size(bin_{(S+f)(i)}) \right] / |bin_{(S+f)(P_1)}|$$

*Proof.* For each of the disjoint partitions of some $S(i) : i \in P_1$, records of the form $i \in P_1$ do not get merged with any records that were not in the initial bin $S(i)$, by definition of partitioning. Thus, summing over all such bins,

$$\left[ \sum_{S(i):i\in P_1} size(bin_{S(i)}) \right] \geq \left[ \sum_{(S+f)(i):i\in P_1} size(bin_{(S+f)(i)}) \right] \tag{11}$$

Note that there is a '$\geq$' inequality since there may be bins in the schema $S+f$ that do contain records of the form $i \in P_1$, which were previously grouped with records $i \in P_1$ in the schema $S$. From Lemma C.3, if the dispersal ratio for population $P_1$ is $r_1$, then $r_1 \geq 1$, which implies

$$|bin_{S(P_1)}| \leq |bin_{(S+f)(P_1)}| \tag{12}$$

Combining equations (1) and (2) proves our result, by observing that they are the numerator and denominator respectively of our desired inequality. $\qquad\square$

Assume two sub-populations $P_0$ and $P_1$ are distributed in the same arbitrary number of bins $|bin_{S(P_1)}| = |bin_{S(P_0)}| = m$. If on adding a feature $f$, $P_0$ and $P_1$ have the same dispersal ratio ($r_0 = r_1 = r'$), then $|bin_{(S+f)(P_1)}| = |bin_{(S+f)(P_0)}| = mr'$. The ratio of their average bin sizes for the schema $S+f$ is

$$\frac{\frac{\left[ \sum_{S+f} size(bin_{(S+f)(i)})_{P_1} \right]}{mr'}}{\frac{\left[ \sum_{S+f} size(bin_{(S+f)(i)})_{P_0} \right]}{mr'}}$$

The average bin size is directly correlated to the size of the sub-population for the same initial number of bins and the same dispersal ratio. Therefore, if one subgroup (say $P_0$) is smaller than the other ($P_1$), then the average bin size for $P_0$ is less than that of $P_1$.

As the average bin size drops for members of a sub-population, the utility will also drop monotonically for partition-based algorithms.

# E   Feature Definitions and Recommended Subsets

Figure 1 lists the 24 Excerpts features. The majority are from the 2019 American Community Survey Public Use Micodata; four of them (DENSITY, INDP_CAT, EDU, PINCP_DECILE) were derived from ACS features or public data as described in C.4. Along with feature type, we've included cardinality (number of possible values). Because some deidentification algorithms require small feature spaces, the NIST CRC program recommends three smaller feature subsets: Demographic-focused, Industry-focused and Family-focused. Each subset showcases different feature mechanics, while sharing common features to delineate subpopulations (SEX, MSP, RAC1P, OWN_RENT, PINCP_DECILE).

| Feature | Description | Type (Size) | In Demog. Feature Set | In Industry Feature Set | In Family Feature Set |
|---------|-------------|-------------|:---:|:---:|:---:|
| PUMA | Geographical area | Categorical (20) | | X | X |
| AGE | Age | Numerical (100) | X | | X |
| SEX | Sex (Male/Female) | Categorical (2) | X | X | X |
| MSP | Marital Status | Categorical (7) | X | X | X |
| HISP | Hispanic Origin | Categorical (5) | | X | |
| RAC1P | Person's Race | Categorical (9) | X | X | X |
| NOC | Number of Own Children | Numerical (11) | | | X |
| NPF | Number of People in Family | Numerical (20) | | | X |
| HOUSING_TYPE | House or Group Quarters | Categorical (3) | X | | |
| OWN_RENT | House Owned or Rented | Categorical (3) | X | X | X |
| DENSITY | Population Density | Categorical (20) | | | |
| INDP | Industry (Work) Code | Categorical (271) | | | |
| INDP_CAT | Industry (Work) Category | Categorical (19) | | X | |
| EDU | Educational Attainment | Categorical (13) | X | X | |
| PINCP | Person's Income | Numerical(1.3M) | | | |
| PINCP_DECILE | Income Decile (by State) | Categorical (11) | X | X | X |
| POVPIP | Income-to-Poverty Ratio | Numerical (502) | | | X |
| DVET | Veteran Service Disability | Categorical (7) | X | | |
| DREM | Cognitive Difficulty | Categorical (3) | | | |
| DPHY | Walking Difficulty | Categorical (3) | | | |
| DEYE | Vision Difficulty | Categorical (3) | X | | |
| DEAR | Hearing Difficulty | Categorical (3) | | | |
| WGTP | Household Sampling Weight | Numerical (1.5K) | | | |
| PWGTP | Person Sampling Weight | Numerical (2K) | | | |

Figure 1: The 24 Features in the Excerpts, and recommended feature subsets.

# F  Detailed Evaluation Reports and Metadata on Selected Deidentified Data Samples

As we noted in the main paper, the NIST CRC Data and Metrics Bundle is an archive of 300 deidentified data samples and evaluation metric results. To demonstrate the efficacy of the Excerpts for identifying and diagnosing behaviors of deidentification algorithms on diverse populations, we selected seven algorithms from the archive to showcase in the paper. Below we provide the complete meta-data and highlighted principal component analysis (PCA) plot for each sample, as well as links to their detailed evaluation reports (available online in the sample report section of the SDNist repository).

Each detailed evaluation report contains the metrics listed below, along with complete results, detailed metric definitions accessible to non-technical stakeholders, a human-readable data dictionary, and additional references.

**SDNist Detailed Report Metrics List:**

- K-marginal Edit Distance

- K-marginal Subsample Equivalent

- K-marginal PUMA-specific Score

- Univariate Distribution Comparison

- Kendall Tau Correlation Differences

- Pearson Pairwise Correlation Differences

- Linear Regression (EDU vs PINCP_DECILE), with Full 16 RACE + SEX Subpopulation Breakdowns

- Propensity Distribution

- Pairwise Principle Component Analysis (Top 5)

- Pairwise PCA (Top 2, with MSP = 'N' highlighting)

- Inconsistencies (Age-based, Work-based, Housing-based)

- Worst Performing PUMA Breakdown (Univariates and Correlations)

- Privacy Evaluation: Unique Exact Match Metric

- Privacy Evaluation: Apparent Match Metric

## F.1    Selected Algorithm Deidentified Data Summary Table

For convenience, we include the deidentified data summary table from the main paper. Expanded results for each algorithm are provided in sections E3-E10)

| Library and Algorithm | Privacy Type | Algorithm Type | Priv. Leak (UEM) | Utility (ES) |
|---|---|---|---|---|
| DP Histogram ($\epsilon = 10$) | differential privacy (DP) | simple histogram | 100 % | $\sim 90$ % (988) |
| R synthpop CART model [2] | non-DP synthetic data | multiple imputation decision tree | 2.54 % | $\sim$40 % (935) |
| MOSTLY AI SDG [3] [4] | non-DP synthetic data | proprietary pre-trained neural network | 0.03 % | $\sim$30 % (921) |
| SmartNoise MST ($\epsilon = 10$) [5] | DP | probabilistic graphical model (PGM) | 13.6 % | $= 10$% (969) |
| SDV CTGAN [6] [7] | non-DP synthetic data | generative adversarial network (GAN) | 0.0 % | $\sim$5 % (775) |
| SmartNoise PACSynth ($\epsilon = 10$) [8] | DP + $k$-anonymity | constraint satisfaction | 0.87 % | $\sim$1 % (551) |
| synthcity ADSGAN [9] [10] | custom noise injection | GAN | 0.0 % | $< 1$% (121) |

Table 1: Summary of selected deidentification algorithms. Unique Exact Match (UEM) is a simple privacy metric that counts the percentage of singleton records in the target that are also present in the deidentified data; these uniquely identifiable individuals leaked through the deidentification process. The Equivalent Subsample (ES) utility metric uses an analogy between deidentification error and sampling error to communicate utility; a score of 5 % indicates the edit distance between the target and deidentified data distributions is similar to the sampling error induced by randomly discarding 95 % of the data. Edit distance is based on the k-marginal metric for sparse distributions. [11], [12]

## F.2 Expanded Data for Subgroup Dispersal Line Graphs

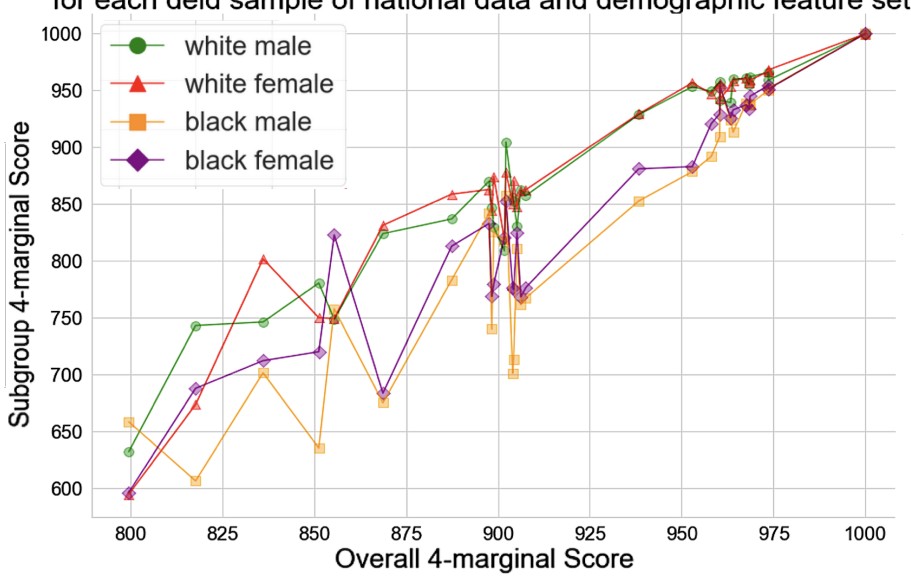

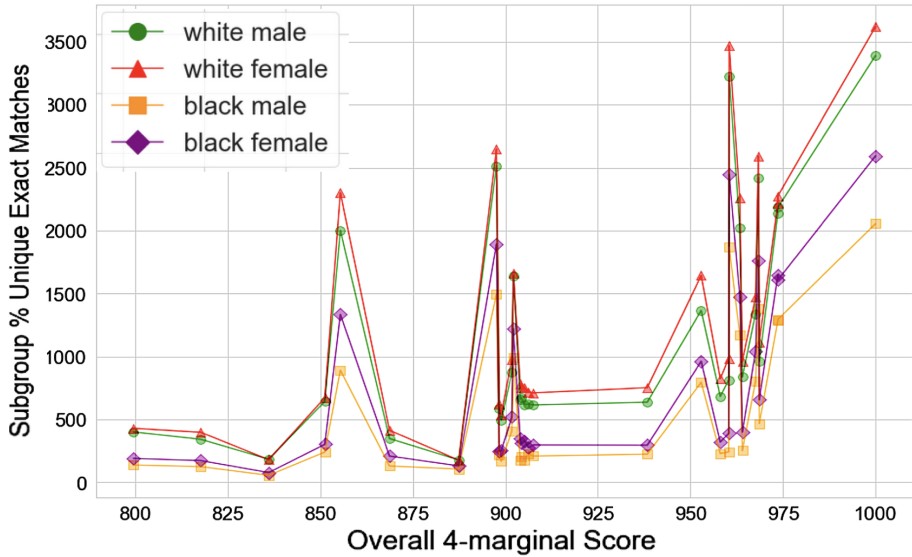

| | library | algorithm | epsilon | subgrp 4-marg. gap | overall UEM | overall percent UEM | overall 4-marg. |
|---|---|---|---|---|---|---|---|
| 0 | rsynthpop | ipf | 1 | 100 | 526 | 3.53 | 836 |
| 1 | subsample_5pcnt | subsample_5pcnt | | 75 | 746 | 5.0 | 887 |
| 2 | rsynthpop | ipf | 2 | 156 | 1223 | 8.2 | 868 |
| 3 | synthcity | tvae | | 64 | 1279 | 8.57 | 799 |
| 4 | smartnoise-synth | pacsynth | 5 | 137 | 1521 | 10.34 | 817 |
| 5 | rsynthpop | ipf | 2 | 95 | 1593 | 10.68 | 898 |
| 6 | smartnoise-synth | mst | 1 | 108 | 1860 | 12.47 | 898 |
| 7 | smartnoise-synth | pacsynth | 10 | 145 | 2013 | 13.49 | 851 |
| 8 | smartnoise-synth | mst | 10 | 101 | 2027 | 13.59 | 906 |
| 9 | smartnoise-synth | mst | 5 | 95 | 2030 | 13.61 | 907 |
| 10 | Sarus SDG | Sarus SDG | 10 | 37 | 2087 | 13.99 | 905 |
| 11 | smartnoise-synth | aim | 1 | 77 | 2105 | 14.11 | 938 |
| 12 | rsynthpop | ipf | 10 | 157 | 2215 | 14.85 | 904 |
| 13 | rsynthpop | ipf | 100 | 155 | 2275 | 15.25 | 904 |
| 14 | rsynthpop | ipf_NonDP | | 58 | 2379 | 15.82 | 958 |
| 15 | smartnoise-synth | aim | 5 | 48 | 2736 | 18.34 | 960 |
| 16 | smartnoise-synth | aim | 10 | 46 | 2789 | 18.7 | 964 |
| 17 | synthcity | bayesian_network | | 12 | 3370 | 22.59 | 901 |
| 18 | rsynthpop | cart | | 25 | 3728 | 24.99 | 968 |
| 19 | rsynthpop | cart | | 78 | 5538 | 37.12 | 952 |
| 20 | subsample_40pcnt | subsample_40pcnt | | 23 | 5963 | 39.97 | 967 |
| 21 | rsynthpop | catall | 10 | 52 | 7066 | 47.37 | 902 |
| 22 | sdcmicro | kanonymity | | 74 | 7073 | 47.41 | 855 |
| 23 | sdcmicro | pram | | 29 | 8690 | 58.25 | 963 |
| 24 | rsynthpop | catall | 100 | 16 | 9351 | 62.68 | 973 |
| 25 | rsynthpop | catall_NonDP | | 17 | 9453 | 63.37 | 973 |
| 26 | sdcmicro | pram | | 22 | 10160 | 68.11 | 968 |
| 27 | rsynthpop | catall | 10 | 38 | 10950 | 73.4 | 897 |
| 28 | sdcmicro | kanonymity | | 11 | 13219 | 88.61 | 960 |
| 29 | rsynthpop | catall | 100 | 1 | 14917 | 99.99 | 999 |

Table 2: This table contains the specific deidentified data samples and metric results used to generate Figures 3 and 4 in the main paper.

\*\*\*\*\*\*\*\*\*\*\*\*\*\*\*\*\*\*\*\*\*\*\*\*\*\*\*\*\*\*\*\*\*\*\*\*\*\*\*\*\*\*\*\*\*\*\*\*\*\*\*\*\*\*\*\*\*\*\*\*\*\*\*\*

## F.3 Differentially Private Histogram (epsilon-10)

A differentially private histogram is a naive solution that simply counts the number of occurrences of each possible record value, and adds noise to the counts. We use the Tumult Analytics library to efficiently produce a DPHistogram with a very large set of bins. Epsilon 10 is a very weak privacy guarantee, and this simple algorithm provides very poor privacy in these conditions. The points in the 'deidentified' PCA are nearly the exact same points as in the target PCA.

The full metric report can be found here.

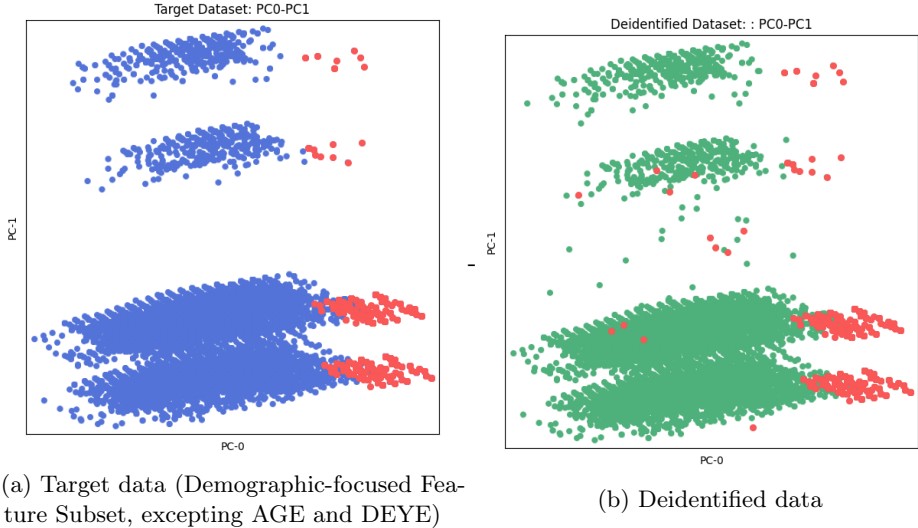

(a) Target data (Demographic-focused Feature Subset, excepting AGE and DEYE)

(b) Deidentified data

Figure 2: The PCA Metric for DP Histogram ($\epsilon = 10$)

| Label Name | Label Value |
|---|---|
| Algorithm Name | DPHist |
| Library | Tumult Analytics |
| Feature Set | demographic-focused-except-AGEP-DEYE |
| Target Dataset | national2019 |
| Epsilon | 10 |
| Privacy | differential privacy |
| Filename | dphist_e_10_cf8_na2019 |
| Records | 27314 |
| Features | 8 |
| Library Link | https://docs.tmlt.dev/analytics/latest/ |

Table 3: Label Information for Differential Private Histogram (epsilon-10)

## F.4   SmartNoise PACSynth (epsilon-10, Industry-focused)

We've included two samples from the PACSynth library to showcase its behavior on different feature subsets. The technique provides both differential privacy and a form of k-anonymity (removing rare outlier records). This provides very good privacy, Table 1, but it can also erase dispersed subpopulations. The industry feature subset below was used for the regression metric in the main paper, which showed erasure of graduate degree holders among both white men and black women.

More information on the technique can be found here. The full metric report can be found here.

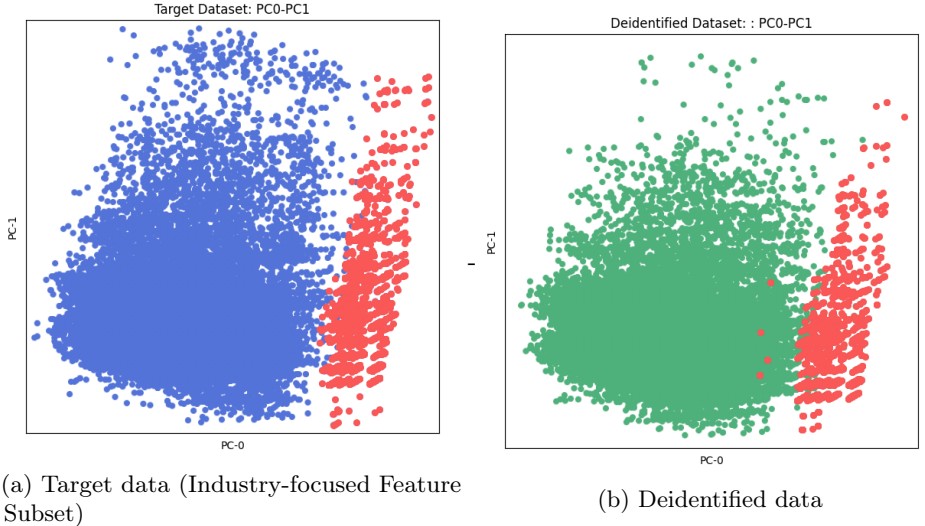

(a) Target data (Industry-focused Feature Subset)

(b) Deidentified data

Figure 3: The PCA Metric for PACSynth ($\epsilon = 10$)

| Label Name | Label Value |
|---|---|
| Algorithm Name | pacsynth |
| Library | smartnoise-synth |
| Feature Set | industry-focused |
| Target Dataset | national2019 |
| Epsilon | 10 |
| Variant Label | preprocessor-epsilon: 3 |
| Privacy | Differential Privacy |
| Filename | pac_synth_e_10_industry_focused_na2019 |
| Records | 29537 |
| Features | 9 |
| Library Link | https://github.com/opendp/smartnoise-sdk/tree/main/synth |

Table 4: SmartNoise PACSynth (epsilon-10)

## F.5    SmartNoise PACSynth (epsilon-10), Family-focused)

On the family-focused feature subset we can see the impact of the k-anonymity protection more dramatically. Because the deidentified data with removed outliers have reduced diversity, it occupies a much smaller area in the plot as compared to the target data. The deidentified records are concentrated into fewer, more popular feature combinations and, thus, their points show less variance along the PCA axes.

More information on the technique can be found here. The full metric report can be found here.

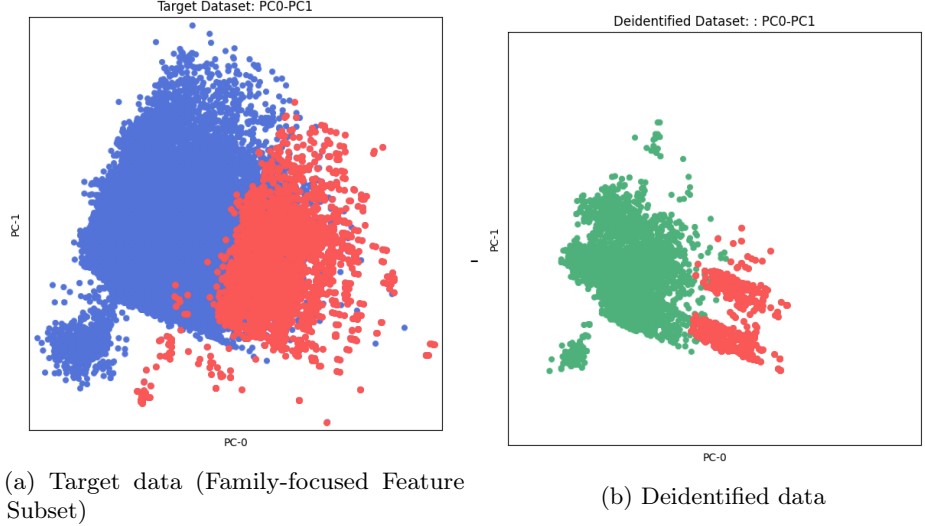

(a) Target data (Family-focused Feature Subset)

(b) Deidentified data

Figure 4: The PCA Metric for PACSynth ($\epsilon = 10$)

| Label Name | Label Value |
|---|---|
| Algorithm Name | pacsynth |
| Library | smartnoise-synth |
| Feature Set | family-focused |
| Target Dataset | national2019 |
| Epsilon | 10 |
| Variant Label | preprocessor-epsilon: 3 |
| Privacy | differential privacy |
| Filename | pac_synth_e_10_industry_focused_na2019 |
| Records | 29537 |
| Features | 9 |
| Library Link | https://github.com/opendp/smartnoise-sdk/tree/main/synth |

Table 5: SmartNoise PACSynth (epsilon-10)

## F.6    SmartNoise MST (epsilon-10)

The MST synthesizer uses a probabilistic graphical model (PGM), with a maximum spanning tree (MST) structure capturing the most significant pair-wise feature correlations in the ground truth data as noisy marginal counts. This solution was the winner of the 2019 NIST Differential Privacy Synthetic Data Challenge. Note that it provides good utility with much better privacy than the simple DP Histogram, but its selected marginals fail to capture some constraints on child records (in red).

More information on the technique can be found here. The full metric report can be found here.

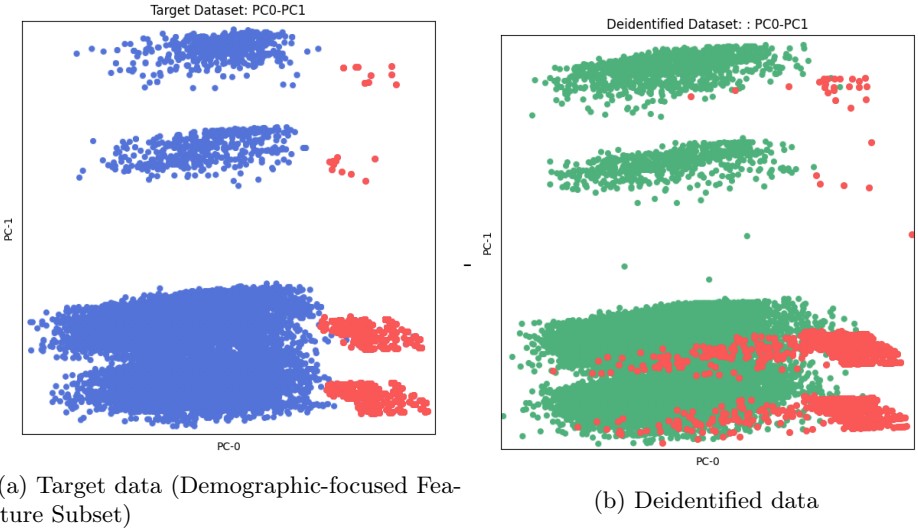

(a) Target data (Demographic-focused Feature Subset)

(b) Deidentified data

Figure 5: The PCA Metric for MST ($\epsilon = 10$)

| Label Name | Label Value |
|---|---|
| Algorithm Name | mst |
| Library | smartnoise-synth |
| Feature Set | demographic-focused |
| Target Dataset | national2019 |
| Epsilon | 10 |
| Variant Label | preprocessor-epsilon: 3 |
| Privacy | differential privacy |
| Filename | mst_e10_demographic_focused_na2019 |
| Records | 27253 |
| Features | 10 |
| Library Link | https://github.com/opendp/smartnoise-sdk/tree/main/synth |

Table 6: Label Information for SmartNoise MST (epsilon-10)

## F.7 R synthpop CART model

The fully conditional Classification and Regression Tree (CART) model does not satisfy formal differential privacy, but provides better privacy than some techniques which do (Table 1). It uses a sequence of decision trees trained on the target data to predict each feature value based on the previously synthesized features; familiarity with decision trees is helpful for tuning this model. Note that the two PCA distributions have very similar shapes, comprised of different points.

You can find more information on the technique here. The full metric report can be found here.

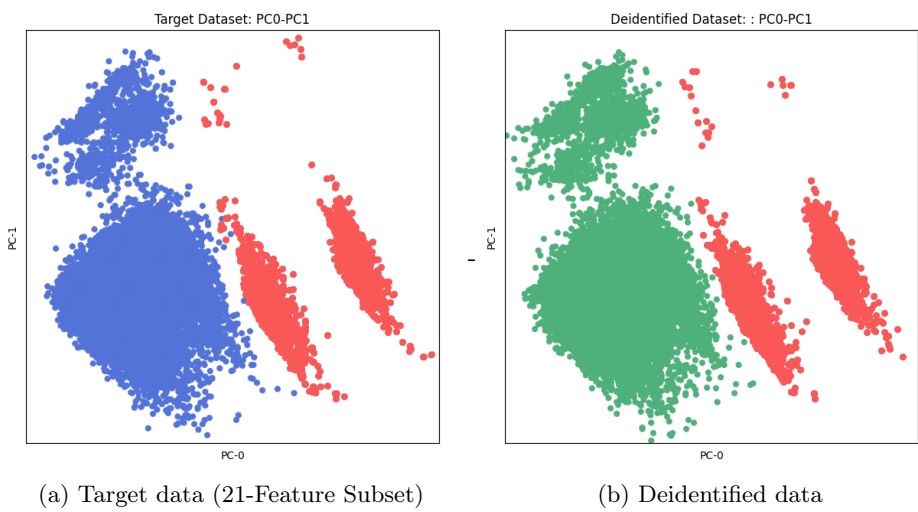

(a) Target data (21-Feature Subset)  (b) Deidentified data

Figure 6: The PCA Metric for CART

| Label Name | Label Value |
| --- | --- |
| Algorithm Name | cart |
| Library | rsynthpop |
| Feature Set | custom-features-21 |
| Target Dataset | national2019 |
| Variant Label | maxfaclevels: 300 |
| Privacy | Synthetic Data (Non-differentially Private) |
| Filename | cart_cf21_na2019 |
| Records | 27253 |
| Features | 21 |
| Library Link | https://cran.r-project.org/web/packages/synthpop/index.html |

Table 7: Label Information for R synthpop CART model

## F.8 MOSTLY AI Synthetic Data Platform

MOSTLYAI is a proprietary synthetic data generation platform which uses a partly pretrained neural network model to generate data. The model can be configured to respect deterministic constraints between features (for a comparison, see MOSTLYAI submissions 1 in the CRC Data and Metrics Bundle linked above). It does not provide differential privacy, but does very well on both privacy and utility metrics (Table 1).

More information on the technique can be found here. The full metric report can be found here.

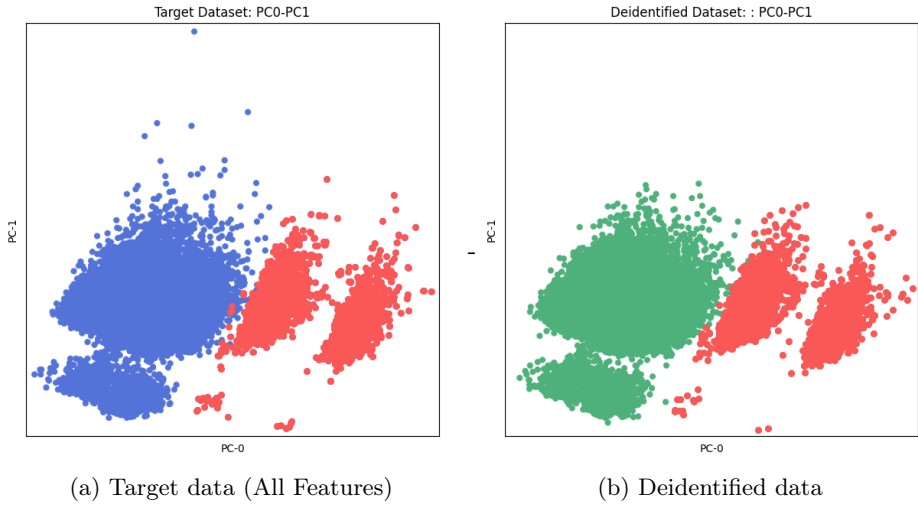

(a) Target data (All Features)      (b) Deidentified data

Figure 7: The PCA Metric for MostlyAI

| Label Name | Label Value |
|---|---|
| Algorithm Name | MOSTLY AI |
| Submission Number | 2 |
| Library | MostlyAI SD |
| Feature Set | all-features |
| Target Dataset | national2019 |
| Variant Label | national2019 |
| Privacy | Synthetic Data (Non-differentially Private) |
| Filename | mostlyai_sd_platform_MichaelPlatzer_2 |
| Records | 27253 |
| Features | 24 |
| Library Link | https://mostly.ai/synthetic-data |

Table 8: Label Information for MOSTLY AI Synthetic Data Platform

## F.9  Synthetic Data Vault CTGAN

CTGAN is a type of Generative Adverserial Network designed to operate well on tabular data. Unlike the MostlyAI neural network (which is pretrained with public data), the CTGAN network is only trained on the target data. It is able to preserve some structure of the target data distribution, but it introduces artifacts. In other metrics, we see it also has difficulty preserving diverse subpopulations.

More information on the technique can be found here. The full metric report can be found here.

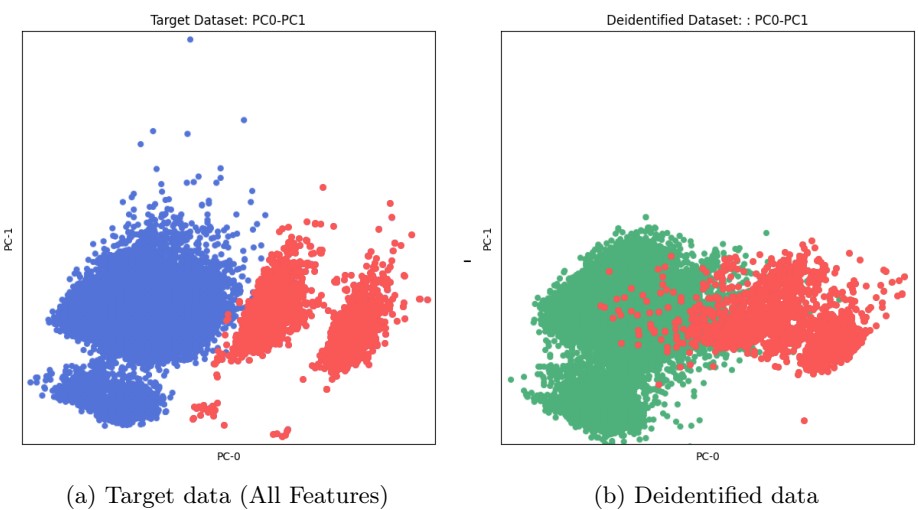

(a) Target data (All Features)   (b) Deidentified data

Figure 8: The PCA Metric for CTGAN

| Label Name | Label Value |
|---|---|
| Team | CBS-NL |
| Algorithm Name | ctgan |
| Submission Timestamp | 4/16/2023 12:03:58 |
| Submission Number | 1 |
| Library | sdv |
| Feature Set | all-features |
| Target Dataset | national2019 |
| Variant Label | default CTGAN with epochs=500 |
| Privacy | Synthetic Data (Non-differentially Private) |
| Filename | sdv_ctgan_epochs500_SlokomManel_1 |
| Records | 27253 |
| Features | 24 |
| Library Link | https://github.com/sdv-dev/CTGAN |

Table 9: Label Information for Synthetic Data Vault CTGAN

## F.10   synthcity ADSGAN

ADSGAN is a Generative Adverserial Network focused on providing strong privacy for synthetic data. While it doesn't formally satisfy differential privacy it uses a parameter alpha to inject noise during the training process. Unfortunately, we see it is unable to preserve any meaningful structure from the target data distribution in this submission.

More information on the technique can be found here. The full metric report can be found here.

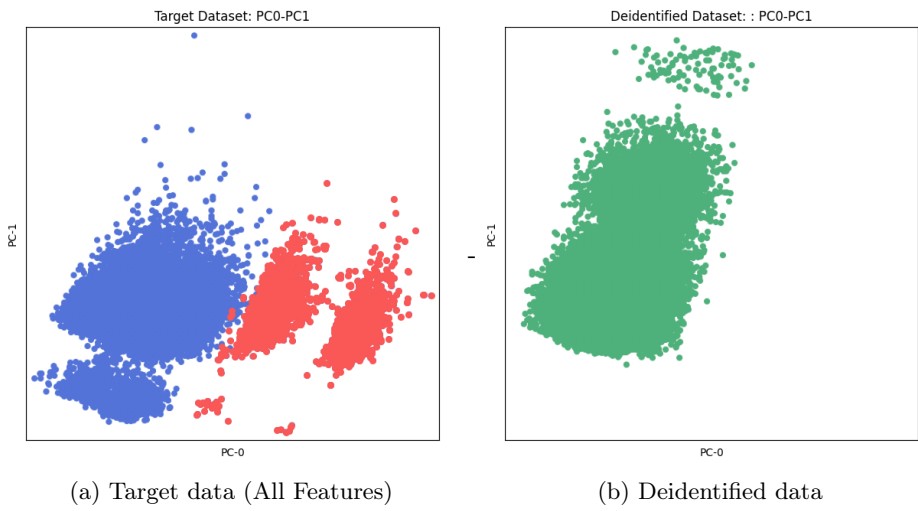

(a) Target data (All Features)       (b) Deidentified data

Figure 9: The PCA Metric for ADSGAN

| Label Name | Label Value |
|---|---|
| Team | CCAIM |
| Submission Timestamp | 3/9/2023 3:33:23 |
| Submission Number | 1 |
| Algorithm Name | adsgan |
| Library | synthcity |
| Feature Set | all-features |
| Target Dataset | national2019 |
| Variant Label | default, lambda=10 |
| Privacy | Synthetic Data (Non-differentially Private) |
| Filename | adsgan_ZhaozhiQian_1 |
| Records | 21802 |
| Features | 24 |
| Library Link | https://github.com/vanderschaarlab/synthcity |

Table 10: Label Information for synthcity ADSGAN