# OpenReview forum: "Diverse Community Data for Benchmarking Data Privacy Algorithms"
_NeurIPS.cc/2023/Track/Datasets_and_Benchmarks — NeurIPS 2023 Datasets and Benchmarks Poster_

### Official Review · Reviewer_efUU · 2023-06-28
**Appears like "2 papers in 1", with main contribution not really fitting datasets & benchmarks track**

**Rating:** 7
**Confidence:** 4
**Correctness:** Basically, the rather isolated contri…

**Strengths:**

- definition and formal analysis of the concept of subpopulation dispersal
- goal of experimentally assessing the quality of generative algorithms for synthetic data is welcome and topical

**Additional Feedback:**

In the current form, the paper appears rather incoherent and undecided about what to focus on, unreflected, and unfitting for this particular track. The different contributions appear quite solid and sound, but at least to me, they don't merge into a consistent story.


Some minor things

* The validity of generative algorithms for deidentification could be put into doubt. Obviously, this approach can only generate datasets that contain (and allow to find) those relations that the generative algorithm recognized in the original dataset first. This hindesr research on potential interrelations not "recognized". Training ML algorithms on so-generated data would need even more explanation: What could a ML-Algo learn from generated data that was not already learned by the initial generative algo? I know, there are arguments in favor of the generative approach here, but this aspect should at least also be discussed in some depth.

* around line 30, it is not clear if it is actually understood what differential privacy does / how ir works. Basically, it is *just not* about releasing the underlying data points but only aggregates thereof.

* line 198: "sex has two categorical values" - might be doubted by some. Why exposing the work without need to? Why not just use sth like "highest educational graduation" or so?

**Clarity:**

The paper is basically well-written, while the overall structure, organization and textual presentation could be significantly improved to support the readers' understanding of the goals that the authors have in mind, of the logical flow of arguments, and of the actual/ultimate contributions and their implications. The fact that the introduction lacks a "the remainder ... structured as follows"-part only reinforces this impression (and adding such a part would not fix the overall problem). In fact, reading the paper requires to "reverse-engineer" this logic etc. in order to understand what the authors want to say.

**Documentation:**

Looks fine

**Ethics:**

no ethical concerns

**Limitations:**

Combining two papers into one also comes at the cost of the non-body parts (especially the discussion of related work and the conclusion) becoming *way* too short and superficial. A conscious discussion of the own contributions, their limitations (except section 3.1), etc. doesn't happen at all.

**Opportunities For Improvement:**

The content part of the paper appears as if it were "two papers in one": One paper on the formal modelling of the concept of subpopulation dispersal (section 2, 4 "body" pages) and one on data excerpts and the experimental evaluation of different deidentification techniques/frameworks (sections 3 and 4, 3 "body" pages) with the formal modelling part being way more elaborate and less superficial. This is in line with the above statement on the main contribution, but raises doubts with regard to the fit of the paper for the datasets and benchmarks track. If the authors had simply published the formal part as a separate paper elsewhere and then concentrated more on the remaining part herein, the whole story would have been more solid, more convincing, and way more fitting to the datasets and benchmarks track.

**Relation To Prior Work:**

The related works section does not mention or discuss any other scientific works pursuing comparable goals and also does not relate/distinguish the own contribution to/from such other works. Instead, the section is just 4 lines of a grouped listing of other tools and libraries.

**Summary And Contributions:**

This paper addresses the challenge of subpopulation diversity / variety in deidentified, publicly released datasets. The subject is of particular importance in the context of anonymous synthetic data created through generative ML algorithms, given the risk of the generated data not properly reflecting the particularities of the underlying "real" data. These risks specifically emanate w.r.t. diverse subpopulations represented in the underlying data.

To address respective challenges, the authors provide a formal definition and analysis of the concept of subpopulation dispersal, identify respective challenges for properly working generative approaches (e.g., related to data-inherent dependencies such as 6-year-old married persons being no valid generation result), software components for analyzing respective properties of datasets, a respective benchmark dataset, and the benchmark results gathered on that basis for different established deidentification techniques. Of these multiple and diverse contributions, the formal definition and analysis is - as stated by the authors themselves as well as from the organization of the paper and the room dedicated to it - the most emphasized one.

---

> ### Author Response · Authors · 2023-08-23
> **Reviewer 4 Response Part 1 (Opportunities for Improvement)**
>
> Thank you very much for your feedback; we provide an itemized response to your primary criticisms below (a discussion of your additional feedback follows).  Please let us know if this addresses your concerns, or if you have any other questions.
>
> ### The content part of the paper appears as if it were "two papers in one" with the formal modelling part being way more elaborate and less superficial.
> - We've significantly shifted the footprint of the two contributions, moving most of the theoretical work to the appendix and expanding the data definition section, as well as adding new sections describing the evaluation tool and data and metric archive.
> - We've also tied the two contributions together more clearly.  The theoretical work exists to answer the question "Why focus the benchmark data on diverse communities?"  This is a key question underlying the need for the project as a whole.  In short, deidentification algorithms can have a range of bad behaviors (bias, artifacts) that makes their use in real world contexts fraught.  To promote research that identifies, diagnoses and addresses these problems we needed benchmark data likely to induce bad behavior  (and comprehensive metrics able to catch it).   Subgroup dispersal is one data property which we can prove arises in diverse populations and which can create a more difficult context for many deidentification algorithms.
> - We've now taken the space to properly document that narrative throughout the paper.  The benchmark data section shows subgroup dispersal in the Excerpts data, and the archive demonstration section shows the empirical performance of deidentification algorithms on the largest four subgroups.  Subgroup dispersal is a stumbling block for some algorithms and not others; understanding what mechanisms underly that is the future research the CRC benchmarking program was designed to promote.
>
> ### The the introduction lacks a "the remainder...."
> - We've now added this, as well as tried to be much clearer about our argument throughout the paper.
> - Informally we wanted benchmark data that would break things. Subgroup dispersal is one key to how things break. It's better to break things in the lab than after they've been put into real world use.
> - *Please let us know if you think our updates have clarified this point sufficiently; we would be happy to adjust more if needed.*
>
> ### The related works section does not mention or discuss any other scientific works pursuing comparable goals and also does not relate/distinguish the own contribution to/from such other works.
> - We've expanded our related works to include more detailed information on comparable benchmarking projects (and we do cite the scientific work for each project, as well as the implemented benchmarks).  We've also more clearly situated ourselves with relation to them.
>
> ### Differential privacy is just not about releasing the underlying data points but only aggregates thereof
> - Thank you for this observation, we've updated that sentence in the introduction to make it more clear that the differentially private techniques produce synthetic data.  This occurs either using neural network models trained with added privacy noise,  statistical models populated by noisy aggregate queries, or other approaches that take in noisy aggregate queries on the target data and produce a collection of new records that mimic those query results. A high level list of algorithm types has been added to the introduction of section 5.  In general we've seen researchers develop differentially private algorithms in nearly every algorithm type category, with clever use of noisy aggregate queries, noisy thresholds, noisy optimization (exponential method), and other sources of randomization.

---

> > ### Author Response · Authors · 2023-08-23
> > **Reviewer 4 Response Part 2 (Discussion of Additional Feedback Points)**
> >
> > ### The validity of generative algorithms for deidentification could be put into doubt
> > - To be frank, I (second author) have at times had doubts about a variety of deidentification techniques.  But rather than try to justify any particular approach, the CRC benchmarking effort exists to document and compare the behavior of these techniques empirically, so those doubts can be explored more rigorously.   Using metrics like Pairwise Correlation, propensity, k-marginal and PCA we can see which feature relationships were retrained, which were lost, and which were altered.  When traditional anonymization techniques provide adequate privacy (non-trivial choices of k and quasi-identifiers) they often lose more than better performing generative techniques, especially for suppression approaches on dispersed subgroups.
> > - Arguably the most important difference between a generative model trained on the original target data and a model trained on the deidentified data, is that the person who made the second model was probably not authorized to see the target data.   I agree that the second model is only getting a diluted version of what was available to (and retained by) the first model, but that's true of deidentification as a whole. Data users have been accepting deidentified data products from government statistical agencies for decades, and any analysis or model training performed on these products will have reduced accuracy that reflects the impact of deidentification.  For example, models trained on data that was deidentified with cell suppression may have worse performance on dispersed subpopulations, because feature correlations for those subgroups were weakened or lost when those individuals' data was suppressed during deidentification. The Diverse Excerpts data, SDNist and the CRC archive enables a search across a wide array of techniques, comparing both anonymization and synthetic data approaches, and it's sometimes surprising how well the best synthetic data techniques are able to retain the distribution of the original data (of course, the worse performing ones can have some very strange impacts on the distribution;  you can see a few examples in our appendix).  Understanding *how and why* some algorithms are successful is something we hope to direct more research attention to.
> > - When using a generative or synthetic data model for deidentification, it is of course possible to release the model itself rather than any particular deidentified data sample, and presumably that would retain even more information.  There's only a couple objections: First if the model wasn't trained/fit with formal privacy (or used a weak privacy parameter) it may be vulnerable to membership inference attacks that aren't feasible on a single deidentified data set.  And second, models are just a bit harder to use than data---data is a nice interface.  Equitable, private, high-fidelity deidentified data is a nice goal  if we can get there, and from some current archive samples (see MostlyAI or AIM) I'm cautiously optimistic we will be able to.
> >
> > ### sex has two categorical values might be doubted by some
> > - Noted, thank you for the catch---We've updated this to refer to vision difficulty, (DEYE)

---

> > > ### Comment · Reviewer_efUU · 2023-08-28
> > > **Way better in most mentioned regards - score raised (without changing text etc)**
> > >
> > > ... thanks for taking into account the remarks so seriously and for actually making such substantive changes to the original version. I think it's way better now (and hope that you also think so).

---

### Official Review · Reviewer_oxgo · 2023-07-02
**Well-written, interesting paper introducing what seems like a rich resource, some room for improvement**

**Rating:** 8
**Confidence:** 3
**Clarity:** Paper is very clearly written.

**Strengths:**

* I rarely say this when reviewing: this paper was mostly easy to follow and pleasant as a reading experience. A lovely clear, motivated introduction, paragraphs that transition well from one to the next, clear definitions of terms and notations, well-explained proofs with both text and equations.
* Contributes theoretically and empirically to a widely relevant topic -- De-identification algorithms and the issue of population dispersal are important, as seen for example in fierce recent debate about differential privacy and the US census. As the authors point out, it is important to understand what kinds of artefacts and biases these algorithms introduce and where in particular they fail. This paper provides tools and resources that could help us do so and fairly effectively demonstrates their use.
* Propose and clearly explain a metric for measuring population dispersal for categorical variables.
* Mostly clear figures and diagrams that are tied into the text and illustrate important points (but see some comments in list below). I especially like concept for Figures 1 and 4
* Well-organized appendix with nice further explanations of dataset and the de-identification algorithms used


**Additional Feedback:**

Some additional notes on things that it would be better to address but don't make or break the paper:

**Medium Notes**
* Figure 1 – took me a moment to understand the boxes (“features? What features?). Perhaps adding to the caption a line like “boxes represent partition of observations by an increasing number of categorical features.” would help. Also, not entirely clear to me what the 55%/44% and  77%/23% represent. By trying out the calculation, I think it’s the % blue vs orange in the ‘non-dispersed,’ non-grey boxes but caption should say that. I do like the concept for this diagram.

* Pg 3 – Figure 2a – I’m inferring that this plot reflects one of the datasets (which?) with dispersal measured as features are added to the model (x axis is cumulative with each feature being added to those on the left)? It would be useful for this to be more explicit in the caption. As noted above, I don’t fully understand what Figure 2b is showing me beyond that some kind of error measured somehow is going up as dispersal goes up.

* Pg. 8 – Figure 4 is a bit blurry with text so small that it is almost impossible to make out. It would be good to increase the resolution and font size of the plot titles and legend. The plot also has some stray marks. Also: I'm confused about why the "Target samples" blue line on top is different across the three plots.  Also, why is the “MOSTLY AI” leftmost plot not discussed in the text, It seems to do well?

**Minor notes**
* Pg. 3 Def 2.2 – “probability distribution of a histogram” is a slightly strange way to put it – probability distributions obtained from a histogram?
* Pg. 3 – typo at bottom of page – should say S and not S'?
* Pg. 4 - in definition 2.3, I can infer $bin_S(P)$  is number of bins containing at least one population unit (so $|bin_S(P)  |≤|S|$ but it would be nice to say this explicitly, esp. since it’s an important point in the proofs. The “set of bins in the histogram corresponding...” is less clear
* Pg. 4 – I would prefer for the statement of theorem 2.3 to define that $f(u)=(1-u)H(X)+H(F)$   with $u=U(X│F)$  immediately so reader doesn’t have to wonder what this is.
* Pg. 5 (top) should the summation in the definition of H(X,F) at the top of the page actually be over Range(X,F) since there could be x in the cartesian product $\script{X}×\script{F}$ with probability 0?
* Pg. 7 I think the data overview should mention the dataset sizes and geographic partitions as on pg. 6 of the appendix




**Correctness:**

Mostly or Entirely – as far as I can tell

---
**Theory**

I have not had time to review the proof of appendix Lemma C.4 but I went through the others and especially those in the main paper and they seem correct...with one hopefully minor exception that I hope the authors can clarify for me:

On pg. 5, equation 5, the first term in the right side of the inequality has a $(|Range(X,F)|-1)(\frac{log|P|}{|P|})) $ and we can drop the second term both because it should be small and because it should be positive, meaning removing it only makes the lower bound less tight. In the next line, the inequality turns into $H(X,F) \geq |Range(X,F)|*(\frac{log|P|}{|P|})$, which is also positive since $|P|>1$. But what happened to the -1 ? Removing this would tend to make the bound tighter?

----
**Empirical**

In terms of the benchmark, as noted above, I am not familiar enough with this area  to evaluate whether the particular de-identification algorithms and metrics used in the demonstration are the most relevant and appropriate for this application. Nothing about the overall approach to curating the benchmark strikes me as suspect.


**Documentation:**

Good

Datasets are well- documented in the appendix, including responses to the Datasheets for Datasets framework. Further information is on GitHub, which is linked to in the paper. Author statements, licensing agreements included.

**Ethics:**

I have no ethical concerns.



**Limitations:**

Re: social impact and risks: the authors fill out a datasheets for datasets checklist in the appendix and describe how the data have already been through the census bureau’s privacy protection process. The current release creates no additional privacy risk. The work is focused on understanding methods that are meant to address privacy risks.

Re: limitations in general: in the main text and appendix, the authors are transparent about some limitations of the current datasets and mention improvements they plan for the future.


**Opportunities For Improvement:**

The paper is split between two quite different parts. Section 2 discusses the issue of subpopulation dispersal and explains it in theoretical detail, with definitions, theorems, and proofs. It is quite self-contained. Sections 3 and 4 describe data and demonstrate comparing various de-identification algorithms on the data using a number of metrics. These are less self-contained, with references to various metrics and algorithms that are not defined in the main text (not saying they should be, but it is a different reading experience). Section 2 takes up a substantial portion of the paper while section 3-4 end up feeling squeezed, with a bit too much left to the appendix and GitHub.

I also feel that the two parts are not tied together quite enough. The introduction provides intuition for why subpopulation dispersal is a problem and illustrates the phenomenon theoretically, but then that theory doesn’t come up in what follows.  The paper proves bounds on dispersal and reasons about their relationship to feature independence, but it could do more to address how a practitioner might apply the theory to think about how to systematically look for weak spots in their de-identification approach – what are we not doing now that we should be? (or they should more explicitly state that they mean the theory to have some separate value)

Relatedly, the paper could do more to tie the theory to empirical demonstration and could discuss dispersal more in relation to the data. For example, I learned only from the appendix that the data include National and Texas samples selected for heterogeneity and a Massachusetts sample selected for less heterogeneity – that seems worth mentioning in the main paper. How could that be useful here? The appendix mentions that the data contain “subpopulations with varying levels of feature independence” (A5) but I don’t see an overview there or in the main paper of what possible subpopulations there are, or which features provide more/less independence relative to others. Again, particularly helpful would be a discussion of takeaways for what people should/might use the dataset to do in this realm that they cannot do with other datasets.

I don’t mean the above to be too harsh – it is still a quite cohesive paper and dispersal is there in the discussion. There is Figure 2, indicating one important axis for dispersal is race [but at the time it is presented, the data for Figure 2a have not yet been explained and I still do not fully understand Figure 2b...what exactly is “error of identification” on the y axis?]. There is Figure 3, which compares a more dispersed and a less dispersed group and with some nice comparative discussion of the algorithms [but re dispersal in particular, I'm unsure what to conclude from the example. The text says the CART and MST create a higher correlation between education and income in the dispersed group, but I don’t actually see that in the plot? I see it in CTGAN, ADSGAN, and PACSynth in the other direction?]. Figure 4 is a nice illustration of algorithms failing on a subpopulation [I’m curious about the authors’ process – was it trial and error to identify that the red clusters corresponded to children? Did they check other subpopulations, too?]

Finally, because sections 3-4 are brief, I am unsure about the nature of the metrics archive and algorithm availability – how many are there? Have these all been applied to the dataset? Or is there code for implementing them that I could transfer to another dataset? Where are they defined? How might I use this resource if I’m apply a de-identification algorithm to my dataset and worried about dispersal? One thing that could help address this is a table in the main text summarizing basic facts about the resources offered and links to where to find more. For the data, metrics, and algorithm resources: what is it? how many? what can you do with it? how do I learn more? The paper links to such a variety of external documentation pages that I found it hard to keep track or know where to begin. Such a table is not a must, but it would be nice.

**Note:** I’m undecided on whether I think the proofs on pg. 5 and 6 are necessary in the main text. If space is a constraint, I think it would be ok for them to be in the appendix (or at least proof of theorem 2.4).


**Relation To Prior Work:**

Minimal

The paper mentions that existing approaches to evaluating de-identification algorithms may miss issues with subpopulation dispersal. Otherwise, the relation to prior work is barely addressed. There is a relevant work section at the very end, but it looks like the authors ran out of space and listed some libraries and tools. I suggest putting section 5 in the appendix and incorporating into the introduction a bit more of an overview of what kinds of benchmarks and approaches are out there and what their limitations are.

**Summary And Contributions:**

The paper addresses introduces resources for studying flaws in de-identification approaches such as differential privacy and synthetic data generation algorithms, particularly in the context of heterogeneous/diverse datasets. The paper focuses on the issue of subpopulation dispersal in which, as more features are added to a dataset, some individuals become increasingly isolated in low-frequency bins (combinations of features), making them particularly susceptible to re-identification and privacy violation. This can impact different subpopulations differently, potentially leading to more adverse impact or error for some groups. Theoretically, the paper defines measure of dispersal and discusses its properties, highlighting its relationship to how independent a new feature is relative to a set of existing ones. Empirically, the paper describes benchmark datasets curated from the American Community Survey with the above purpose in mind. It then demonstrates some comparisons of algorithms on these data using a number of metrics that highlight different aspects of performance.

Note: deidentification algorithms are not my particular area of expertise; this review reflects how well an interested, statistically literate academic with some but not extensive exposure to differential privacy and only very light exposure to synthetic data generating algorithms can understand the paper.

---

> ### Author Response · Authors · 2023-08-23
> **Reviewer 3 Placeholder**
>
> We sincerely thank reviewer 3 for their very detailed feedback; we hope the most significant concerns were address by the general response on Monday.  We also have worked to address the more fine-grained concerns as well, and are taking one more day to complete our response.

---

> > ### Comment · Reviewer_oxgo · 2023-08-27
> >
> > Thank you for taking into account my suggestions. I think the new version of the paper is improved, with better explanations of the data and tool contributions. The paper still tries to do a lot of different things but the separate contributions are clearer and more balanced. The figures are better explained and figure placement is better.

---

### Official Review · Reviewer_wNKf · 2023-07-19
**Diverse Communities Data Excerpts, partnership with SDNist and theory to address subpopulations**

**Rating:** 7
**Confidence:** 4
**Correctness:** Yes
**Clarity:** There's room for improvement, see "Op…

**Strengths:**

This paper presents four notable contributions.

Firstly, it introduces new theoretical insights into the relationship between diverse populations and the challenges associated with equitable identification.

Secondly, it provides a valuable public benchmark dataset that specifically focuses on diverse populations and includes challenging features extracted from the American Community Survey.

Thirdly, it offers an open-source suite of evaluation metrics designed for assessing the quality of deidentified datasets.

Lastly, it presents an extensive archive of evaluation results, encompassing a wide range of deidentification techniques.

**Additional Feedback:**

None

**Documentation:**

Yes

**Limitations:**

As pointed out by the authors, one limitation of the Excerpts approach is the absence of Household IDs, which hinders the synthesis of social networks within households. Additionally, the lack of Individual IDs restricts research on reidentification. Another limitation is the absence of a clear training/testing partition, crucial for differential privacy research.

**Opportunities For Improvement:**

In Section 2 of the paper, the concepts presented can be considered quite abstract. Notably, the proofs for Theorem 2.3 and Theorem 2.4 are extensive, taking up two pages. To enhance the readability and flow of the main text, I recommend relocating these proofs to the appendix. In their place, incorporating easy-to-understand examples would be beneficial to illustrate the impact of subpopulations. These examples will not only aid in comprehending the theorems but also highlight the necessity for tools that specifically address subpopulation challenges.

**Relation To Prior Work:**

Section 5 of the paper introduces related works; however, the discussion regarding how the proposed method differs from previous approaches is concise and lacking in depth. To address this concern, a more comprehensive analysis and comparison between the proposed method and existing works should be provided.

**Summary And Contributions:**

This paper presents the Diverse Communities Data Excerpts, developed in partnership with SDNist and theory, as a novel solution to address the persistent problem of deidentification technologies in diverse subpopulations.

---

> ### Author Response · Authors · 2023-08-23
> **Reviewer 2 Response**
>
> Thank you very much for your feedback; we provide an itemized response below.
>
> ### In Section 2 of the paper, the concepts presented can be considered quite abstract. Notably, the proofs for Theorem 2.3 and Theorem 2.4 are extensive, taking up two pages.
> - This was a common complaint, thank you for the suggestion.  We've moved the proof work to the appendix, and expanded our intuitive explanations of dispersal in the introduction and at the beginning of section 2 to make this material more readily accessible.
>
> ### A more comprehensive analysis and comparison between the proposed method and existing works should be provided.
> - In table 2 we now provide detailed information about the metrology and assets available in all of our comparable deidentified data benchmarking libraries.  Additionally, in the related works section and in the introduction, we've better clarified our position with respect to them.

---

> > ### Comment · Reviewer_wNKf · 2023-08-23
> >
> > Thank you for addressing my concerns, I've raised the score.

---

### Official Review · Reviewer_aPv5 · 2023-07-23

**Rating:** 5
**Confidence:** 3

**Strengths:**

- The paper introduced a benchmark for comparing the deidentification algorithms on diverse subpopulations, which provides a pathway for a better understanding of the tradeoff between privacy and fairness.

- The paper presented a good combination of theoretical efforts and empirical results. In particular, the paper presented some interesting results showing how the dispersal of a population would change with additional features.



**Additional Feedback:**

None.

**Clarity:**

The paper is in general well written but can benefit from improvements in the following aspects.
- The notation system could be improved for better clarity. E.g., Range(X,F) in line 92 is undefined.
- It is unclear the connection between the theory section (Section 2) and the following sections.

**Correctness:**

The main concern about the paper is how to rigorously compare different deidentification algorithms that offer fundamentally different privacy guarantees. For example, the paper has claimed that "MST is a good compromise of privacy and utility (and would provide better privacy at smaller ε), but non-differentially private techniques CART and MostlyAI are better on both privacy and utility." It is unclear how to compare the privacy strength of differentially private vs. non-differentially private algorithms.

**Documentation:**

The documentation is well-written.

**Ethics:**

No.

**Limitations:**

The authors have addressed the limitations of their work in Section 3.1.


**Opportunities For Improvement:**

- The connection between the theory section and the empirical evaluation is not clear. How could the theory benefit empirical evaluation?

- The paper can benefit from a detailed discussion of how to compare different algorithms that offer different privacy guarantees.

- The dispersal metric (the ratio of the number of bins before and after adding a feature) could be better justified. This metric overlooks how the population distributions within the bins and just count the change in the bin number. Why is it considered a useful metric for dispersal? How does it compare to other potential metrics like change in variance?




**Relation To Prior Work:**

The difference from previous work is not sufficiently discussed. It is recommended to expand the related work section and compare the similarity and differences between existing benchmarks and libraries.

**Summary And Contributions:**

The paper provides a benchmark for comparing different data deidentification algorithms. The paper makes the following contributions: (1) it introduced the concept of dispersal ratio and proved several results on the relationship between dispersal and feature dependence. (2) It introduced public benchmark data focused on diverse populations. (3) It presented evaluation results on some existing deidentification techniques.

---

> ### Author Response · Authors · 2023-08-23
> **Reviewer 1 Response**
>
> Thank you very much for your feedback; we provide an itemized response
>
> ### The connection between the theory section and the empirical evaluation is not clear.
> - We've strengthened the connection of the concept of dispersal to our data excerpts by adding section 5 (The Collaborative Research Cycle Data and Metrics Archive) and with the addition of two new analyses illustrating subgraph dispersal in the Benchmark Data (figure 3) and Archive (figure 4)
> - The theory underlies the choice to focus our benchmark data and the CRC program as a whole on diverse communities.  By robustly evaluating in more challenging real world conditions, we hope to help algorithm developers identify problems they might overlook with simpler test data or metrics.  We've attempted to clarify this in the introduction, *please let us know if you think we were successful.*
>
> ### It is unclear how to compare the privacy strength of differentially private vs. non-differentially private algorithms.
> - We've extended discussion of the evaluations with special attention to the privacy evaluation and the UEM metric in Section 4 (see final subsection for UEM specifically), and improved references in the demonstration section.
> - A primary difficulty in considering formal privacy guarantees is that in real world practice people often run formally private techniques at parameter settings that don't guarantee privacy.  When epsilon = 10, differential privacy actually permits the output to be over 22,000 times more likely to have come from the real world as any neighboring world (when relative probability is bound by e^10, it's not especially bound).  This is how the DP histogram is able to nearly exactly reproduce the input without violating its guarantee  (check Appendix E.3 for an alarming visualization). High epsilons don't generally cause outcry in real world usage because DP techniques often include other untracked sources of randomness or information loss that, in practice, add up to a reasonable amount of obfuscation (See E.6 for comparison).  But in that respect they become more comparable to non-DP techniques, which also integrate untracked randomness and information loss for privacy's sake.
> - To make a reasonable comparison about what's actually happening once these algorithms run on real data (rather than what a potentially weak guarantee will allow to happen; relatively few DP algorithms will provide usable fidelity at epsilons below 2, even for our smaller feature sets), we've resorted to a simple reidentification test.  If a record was unique under a given feature set in the selected target data, we consider it potentially vulnerable to reidentification.  If that exact same record appears unaltered in the deidentified data, we hypothesize that reidentification might not be difficult.  As we mentioned in the global review response and in the updated paper, this naive approach has clear limitations.  But if a privacy technique has a Unique Exact Match (UEM) score of 30%, then a more robust reidentification attack is unlikely to return fewer reidentified records.   We consider UEM to be a lower bound on empirical privacy leakage.
> - There are more sophisticated ways to compare privacy across different definitions/approaches, and we hope to explore those in future versions of SDNist; the Anonos library cited in our related works table is a good example.  But in the current version of SDNist, if the privacy provided by combined tracked and untracked randomness in a differentially private data generator results in more UEM leakage than the privacy provided by untracked randomness in a non-dp synthetic data generator, then we're able to document that as a point relevant for discussion.
> - *please let us know whether you feel this answers this question adequately.*
>
> ### The dispersal metric (the ratio of the number of bins before and after adding a feature) could be better justified.
> - We've added further explanation after the definition is introduced in section 2.  Thank you for the observation; a point we hadn't made clear before is that our data (and most tabular survey data) is largely comprised of non-ordinal, categorical features (i.e., the answers to multiple choice questions). For a sub-population of a fixed size, this definition of dispersal ratio captures the increase in the number of histogram bins the population is distributed across on when a new feature is introduced, resulting in a reduction of the average count per bin and a monotonic increase in the number of small-count bins.  We discuss earlier in section 2 the implications of small count bins for a variety of privacy approaches.
>
> ### The difference from previous work is not sufficiently discussed.
> - In the introduction we attempt to distinguish this project from related work. We've expanded our survey of synthetic data benchmarking libraries to include details of the comparable libraries' metrology and assets, and clarified our position with respect to them.

---

### Author Response · Authors · 2023-08-21
**Initial Global Review Response (Paper update forthcoming)**

Our sincere thanks to the reviewers for providing this feedback on our work.

We are completing work on our second draft, which we plan to upload Tuesday, 22 Aug. However, we wanted to provide you with a list of updates to expect and initial responses to your feedback.

These issues were commonly identified across our reviews:
* *Division between theory and benchmark sections.*
* *Too much space dedicated to theory section*
* *Too little space dedicated to the data/benchmark section*
* *Unclear handling of privacy evaluations (especially given different privacy definitions/philosophies)*
* *Related works needs expansion to compare to other tools*
* *Other observations/questions/unclear points*


We have worked to resolve these issues in the following ways:
* **Division between theory and benchmark sections:** We've added analyses on the CRC Data and Metrics Archive to illustrate the subgroup dispersal in the benchmark data, and its empirical impact on deidentification algorithm performance across the archive.
* **Too much space dedicated to the theory section:** We left the main conclusions in the body and relegated the proof work to the appendix.
* **Too little space dedicated to the data/benchmark section:** We've expanded the benchmark data description section, and added new detailed sections covering the metric library and the data and metrics archive as a whole. These sections now also include demonstrations of the diverse subpopulation problem our benchmarking data and metrics hopes to help address.
* **Unclear handling of privacy evaluations (especially given comparisons across differing privacy definitions/philosophies)**: With our new metrics section, we give proper space to describing the empirical UEM privacy metric, our reasoning behind it, and how it relates to the results discussed in Table 1.  In essence, we look at a simple but unambiguous privacy leak metric that gives us an empirical lower bound on privacy.  With this we can make a reasonably clear argument that one algorithm is performing significantly worse than another on privacy, even though the metric may be insufficient to support the claim that any particular algorithm is performing ``well’’, or distinguish more subtle differences in performance.  We hope to expand on our privacy metric section in future releases of the library, but as a starting point this simple empirical check provides some occasionally surprising insights.  We also include privacy guarantee parameters for any method that has them.
* **Related works too short**: We've added a chart showing a detailed breakdown and comparison of the metrics and assets available in all of the libraries we linked previously. In the introduction we provide better context of where our project situates itself within the field.
* **Other observations/questions/unclear points**: We're addressing the other reviewer comments as well and will note these in our reviewer-specific responses.

---

### Decision · Program_Chairs · 2023-09-22

**Decision:**

Accept (Poster)

**Comment:**

This paper studies tabular data deidentification technologies, which are vulnerable to bias and privacy issues with several contributions: introduced the concept of dispersal ratio and proved several results on the relationship between dispersal and feature dependenc; public benchmark data focused on diverse populations and challenging features; an open-source suite of evaluation metrics designed for assessing the quality of de-identified dataset; and an archive of evaluation results, encompassing a wide range of deidentification techniques.

The initial reviews are mixed. Several reviewers raised the concerns on the division of theory and benchmark sections, the lack of connection to previous related works, and unclearness of privacy guarantees and metrics. The authors substantially changed the presentation and made several successful clarifications of the paper, which resulted in score raising. The reviewers generally acknowledged the contributions after the discussion phase.

Privacy is an emerging and important topic in machine learning. High quality dataset with benchmark algorithms can inspire future studies. I personally found the discussion for this paper inspiring. Thank you for the solid submission!